

# The Alaiz Experiment: untangling multi-scale stratified flows over complex terrain

Pedro Santos[1], Jakob Mann[1], Nikola Vasiljević[1], Elena Cantero[2], Javier Sanz Rodrigo[2], Fernando Borbón[2], Daniel Martínez-Villagrasa[3], Belén Martí[3], and Joan Cuxart[3]

[1]Technical University of Denmark, DTU Wind Energy, Risø Campus, Roskilde, Denmark
[2]National Renewable Energy Centre (CENER), Sarriguren, Spain
[3]Physics Department, University of the Balearic Islands (UIB), Mallorca, Spain

**Correspondence:** Pedro Santos (paas@dtu.dk)

**Abstract.** We present novel measurements from a field campaign that aims to characterize multi-scale flow patterns, ranging from 0.1 to 10 km, in a mountainous region in Northwestern Spain with a mountain-valley-ridge configuration. We select two flow cases where topographic-flow interactions were measured by five synchronized scanning Doppler wind lidars along a 10-km transect line, including a cross-section of the valley flow. We observed a hydraulic jump in the lee-side of the mountain.
The Froude number transition from supercritical ($> 1$) at the mountain to subcritical ($< 1$) at the valley is in agreement with previous experiments at a smaller scale. For a one-year period, the measurements show such a transition about 10 % of the time, indicating a possible high occurrence of hydraulic jumps. The second flow case presents valley winds that are decoupled from the northerly flow aloft and show a stratified layered pattern, which is well captured by the lidar scans. These measurements can aid the evaluation of multi-scale numerical models as well as improving our knowledge with regards to mountain meteorology.

## 1 Introduction

Over flat and homogeneous terrain, such as areas far offshore, the difference between measured and simulated mean wind speeds is rather low at wind energy relevant heights, in some cases less than 4% (Olsen et al., 2017). However, over complex terrain, with steep slopes and varying land cover, such differences can be more substantial, implying large uncertainties on the estimated annual energy production of wind farms. Even small deviations in the terrain description over a given area may result
in substantial differences in the simulated flow (Lange et al., 2017).

For the prediction of winds in complex terrain, meso-scale models, typically covering scales down to a kilometer or so, have to be coupled with micro-scale models that cover smaller scales down to meters. Meso- and micro-scale models are fundamentally different in the sense that flow processes that are parameterized in the former are resolved in the latter, while several physical processes are included in the former but not in the latter. The scales that are at the interface of the two models
have been dubbed *terra incognita* by Wyngaard (2004) and this experimental investigation aims to explore some sub-meso-scale physical processes. New datasets from complex terrain experiments with details on flow patterns covering these scales are rare and needed to evaluate and quantify the uncertainty of numerical models (Sanz Rodrigo et al., 2017). Apart from wind





energy, untangling flow over complex terrain is of general interest for the mountain meteorology community (Serafin et al., 2018).

Over the last decades, experimental efforts have been conducted with increasing density of instruments and types of measurement aiming to better understand flow conditions in hilly and mountainous terrain. A well-known experiment was performed at

the Askervein Hill, which became the main reference in the development and validation of pioneering analytical and linearized flow models dealing with gently sloping terrain (Salmon et al., 1988; Walmsley and Taylor, 1995). Furthermore, the Cooper's Hill experiment (Coppin et al., 1994) used meteorological masts and sonic anemometers to study the flow over a ridge as a function of atmospheric stability.

In more recent endeavors, Doppler wind lidars and airborne instrumentation have been used to characterize large-scale phe-

nomena over steep hills and mountain ranges. Two examples of such are the terrain-induced rotor experiment (T-REX, Grubišić et al., 2008) and the mountain terrain atmospheric modeling and observations program (MATERHORN, Fernando et al., 2015). T-REX focused on low-level vortices formed downstream of a mountain ridge and MATERHORN was a multidisciplinary initiative to approach large-scale atmospheric phenomena in complex terrain, where two major experimental campaigns studied thermally driven winds with strong synoptic forcing. Back to smaller scales, detailed scanning lidar and turbulence measure-

ments were performed at the escarpment of Bolund (Lange et al., 2016; Berg et al., 2011), which brought new insights into flow-terrain interaction. A blind test followed to compare a wide variety of flow models (Bechmann et al., 2011) and wind tunnel prototypes (Kilpatrick et al., 2016; Conan et al., 2016; Lange et al., 2017).

With an extensive collaboration effort in the pursuit of new insights on wind resource characterization, a range of experiments, both onshore and offshore, were performed within the New European Wind Atlas (NEWA) project to evaluate meso-

and micro-scale models (Mann et al., 2017). The experiments made extensive use of a recently developed infrastructure that uses synchronized measurements from multiple lidars, the so-called long-range WindScanner system (Vasiljević et al., 2016). The Kassel experiment, performed at the forested hill Rödeser Berg in Germany, was used to quantify the accuracy in the reconstruction of the wind vector with distinct multi-lidar combinations and the lidar's spatial averaging effect on the turbulence spectra (Pauscher et al., 2016). A methodology for the execution of experiments involving multi-lidars was developed

during the double-ridge Perdigão experiment in Portugal (Vasiljević et al., 2017), which is the largest experimental venture in complex terrain to date in terms of density of measurement equipment (Fernando et al., 2019). In parallel to the NEWA project, the second Wind Forecast Improvement Project (WFIP2), also deployed a large array of instruments to cover the area around the Columbia Gorge in the United States (Wilczak et al., 2019). This experiment was also focused on the improvement of meso- and micro-scale coupling methods (Haupt et al., 2019).

Figure 1 puts the Alaiz experiment in perspective of these complex terrain experiments by comparing the area covered and steepness of the terrain quantified by the upper 10th percentile of the slopes. In this context, Askervein can be seen as a departure point that gave rise to experiments in larger areas and steeper slopes.

With a small domain but a steep escarpment, Bolund left the realm of gentle slopes, and hence emphasized the limitations of linearized flow models. Due to the very small scales, neglecting the effects of atmospheric stability did not lead to major

errors in Bolund. On the other hand, in METCRAX II, scanning lidars captured atmospheric hydraulic jumps and cool pool





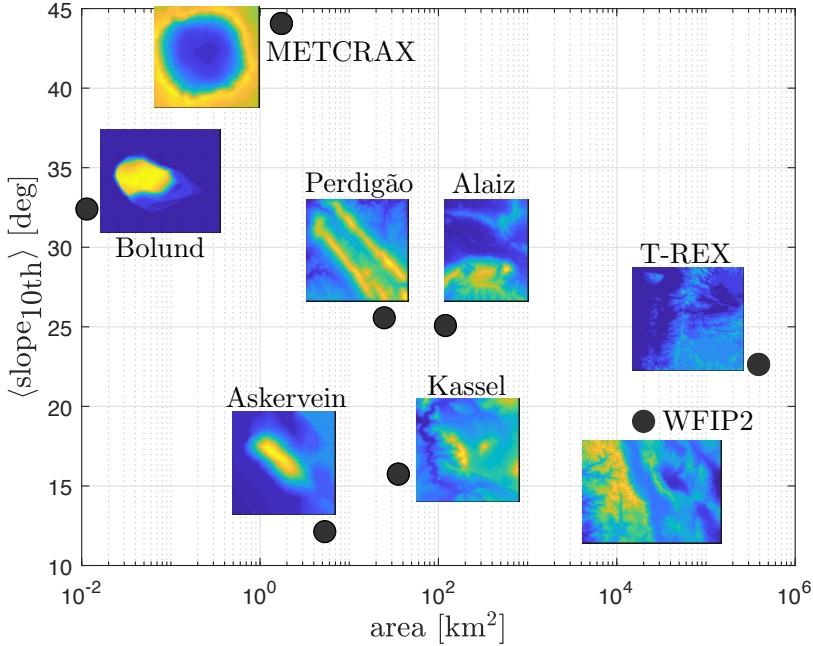

**Figure 1.** Illustration of the complexity of some atmospheric flow experiments as a function of their area of coverage. The complexity is in terms of the slope: the average 10th percentile of the highest slopes. The inserts besides the markers show contour maps of the relative elevation for each site.

events inside a meteorite crater in Arizona (Lehner et al., 2016). In Kassel and Perdigão, larger areas were investigated that required the use of long-range WindScanners. Perdigão presents a double-hill configuration, 1.5 km apart, which is dominated by micro-scale effects, such as valley winds and recirculation zones, but is also affected by thermal stratification effects that can lead to internal atmospheric gravity waves under stable conditions (Menke et al., 2019; Palma et al., 2019). T-REX and

5 WFIP2 are mountain range studies, too large to be fully covered by a single set of instruments, still with similar thermally stratified flows presented in this study.

As highlighted by Mann et al. (2017), Alaiz covers the mid-range where both micro-scale and meso-scale effects are prevalent. As shown in Figure 1, Alaiz features similar complexity as Perdigão, but is an order of magnitude larger with influence of topographic features that are 10 km or more apart. The area to cover is still within the range of current commercial wind lidars.

This paper presents the Alaiz Experiment (ALEX17), which aims to peer into multi-scale flow patterns with mountain-valley interactions. In this work, we are describing two types of flow cases: a layered-stratified valley flow and a hydraulic jump, characterized with multi-lidar measurements and multiple ground-based observations.

This paper is outlined as follows. In section 2 we describe the experimental layout and detail the measurement equipment. Section 3 describes the site climatology and the atmospheric stability is assessed. In section 4 we characterize and discuss the

15 two selected flow cases. Section 5 summarizes the main findings and promotes this data collection as a tool for further analysis and numerical model evaluation.





## 2 The ALEX17 experiment

### 2.1 Site Characterization

ALEX17 took place in the Navarre region, in the northwestern part of Spain. The experimental area encompasses the Alaiz mountain range, a region at 1000 m above the mean sea level (asl) with a wind regime favorable for wind energy applications

5 (Sanz Rodrigo et al., 2013). Figure 2a shows the experimental domain (yellow square) within the Iberian Peninsula with a 1 arcsec resolution elevation map from the Shuttle Radar Topographic Mission (SRTM). The site is situated to the northwest of the Ebro valley, a river basin enclosed by the Pyrenees to the north and the Iberian system to the south.

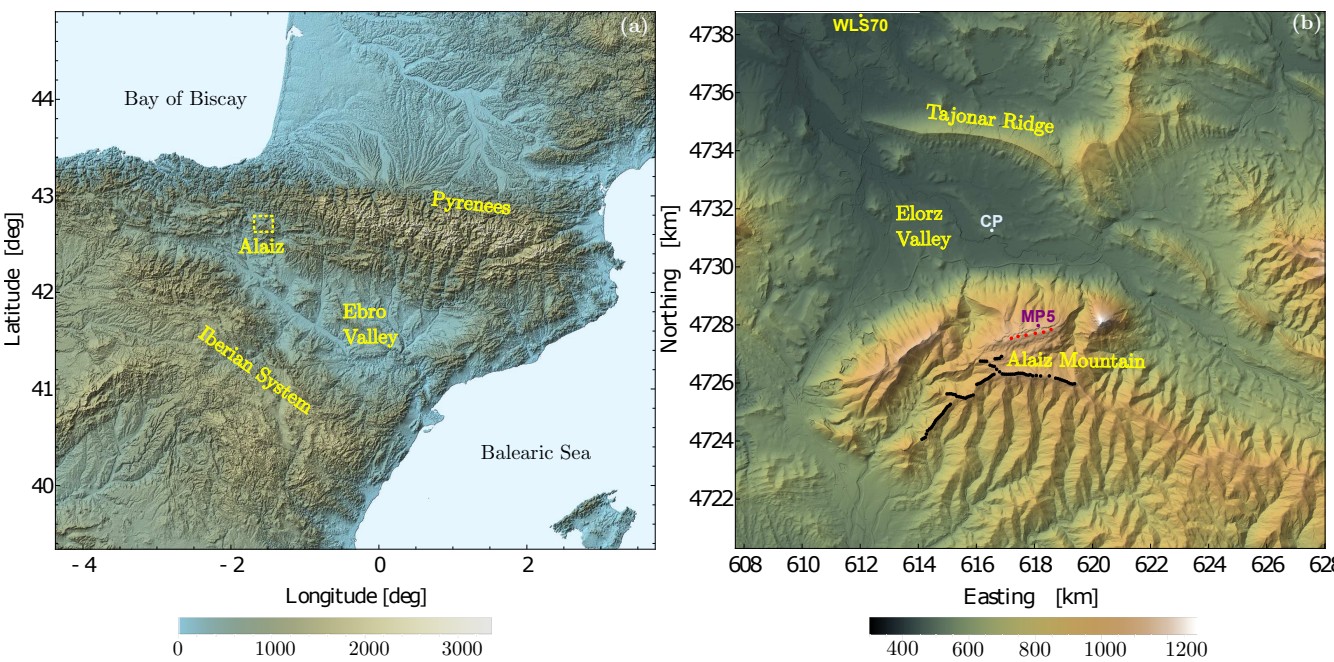

**Figure 2.** Location and overview of ALEX17. Panel **(a)** shows the experiment location (yellow rectangle) within the Iberian Peninsula. The experiment is shown in **(b)**, with: CENER's wind turbine test stands (red) and reference MP5 meteorological mast (purple); Acciona's wind farms (black); the central position (CP, light blue) and a profiling wind lidar (WLS70, yellow). The colorbars represent the height above mean sea level in meters based on digital elevation models from SRTM (panel a) and lidar aerial scans (panel b) in UTM30 WGS84.

The large-scale topographical features in the region explain the synoptic forcing present on this site. Jiménez et al. (2013) performed meso-scale modeling with 2-km horizontal resolution over 45 yr and assessed the wind variability over the region.

10 Badger et al. (2014) presented a statistical-dynamical downscaling to estimate a generalized wind climate in the same area. Apart from inherent biases between model and observations, both studies showed two main circulation patterns, with northwest (NW) and southeast (SE) flow over the Ebro valley, with a channeling effect caused by an orographic funnel formed by the large-scale features (see figure 2a). The NW circulation is ordinarily called "Cierzo". Badger et al. (2014) additionally showed





that this effect is intensified during stable conditions, where the stratified atmospheric boundary layer (ABL) interacts with the orography more actively.

Figure 2b shows the area surrounding the experimental domain with a 2x2-m resolution terrain elevation map based on airborne lidar scans. The Alaiz mountain, to the south, hosts CENER's wind turbine test site in complex terrain with six test
stands (red dots) and a 118-m meteorological mast called MP5 (purple dot). The other test site's masts (not shown) are not part of this experiment. To the south of the mountain plateau, 89 wind turbines (black dots) belong to Acciona's wind farms called Alaiz and Echagüe, which shows that the site, although challenging, is attractive for wind energy production.

To the north of the Alaiz mountain range, most of the measurement equipment are located at the Elorz valley, 500 m asl, around the central position (CP, blue dot) of the experimental domain. The valley is roughly 10 km wide and 6 km long, bounded
by the Tajonar ridge, hereafter called north ridge, which has its peak at 850 m asl. Notice that, as in the large-scale features, the Alaiz mountain and the north ridge are not parallel, also shaping the valley as a funnel. The land cover is heterogeneous (c.f. Figure 4 in Cantero et al., 2019) with villages and distinct kinds of farmland distributed along the valley floor, as well as forest patches on the slopes of the north ridge and the top of Alaiz mountain.

## 2.2 Timeline and Instrumentation

The extensive measurement period (EMP) ran from July 2017 to July 2019, comprising two full years of measurements from the reference mast MP5 and from a long-range profiling wind lidar (WLS70) located at the north boundary of the domain (see figure 2b). The Intensive Observational Period (IOP), when all sensors had concurrent measurements, lasted for five months from August 2018 to December 2018.

Figure 3 shows the 100-km$^2$ experimental domain, with CP at the center and details of the instrumentation layout. Most in-
struments were placed in the valley floor, aiming to investigate the topographic interaction on the flow between Alaiz mountain and the north ridge in order to characterize the wind regime at the mountain top, regarded as the region of interest for wind energy.

**Reference meteorological mast (MP5).** This is a 118-m meteorological mast located at the mountain top (42°41.7′ N, 1°33.5′ W). We selected this mast as a reference because it is measuring continuously since 2011 with the same configuration.
The wind speed and wind direction data availability at 118 m was 85.4 % between February 2011–January 2019. The mast is equipped with wind and temperature measurements distributed in six main levels: 2, 40, 80, 90, 100 and 118 m above the ground level (agl).

**Long-Range WindScanners (WS).** Five WindScanner units were distributed along the valley floor, depicted with yellow markers in figure 3. By scanning on vertical planes across the valley, it is possible to visualize the dynamics of multi-scale
flow patterns generated by the interaction between the ABL and the topography such as atmospheric waves or flow-separation regions in the lee side of the ridges. We configured the units to synchronously scan in pairs and triplets, allowing the wind vector reconstruction on top of the ridges and across the valley, shown by red and blue lines in figure 3 and detailed in Sec. 2.3.

**Meteorological masts (M).** Six 80-m tall meteorological masts were distributed across the valley, marked with distinct colors in figure 3 for masts with either cup/vanes or 3D sonic anemometers. They are located either on the valley floor or in the





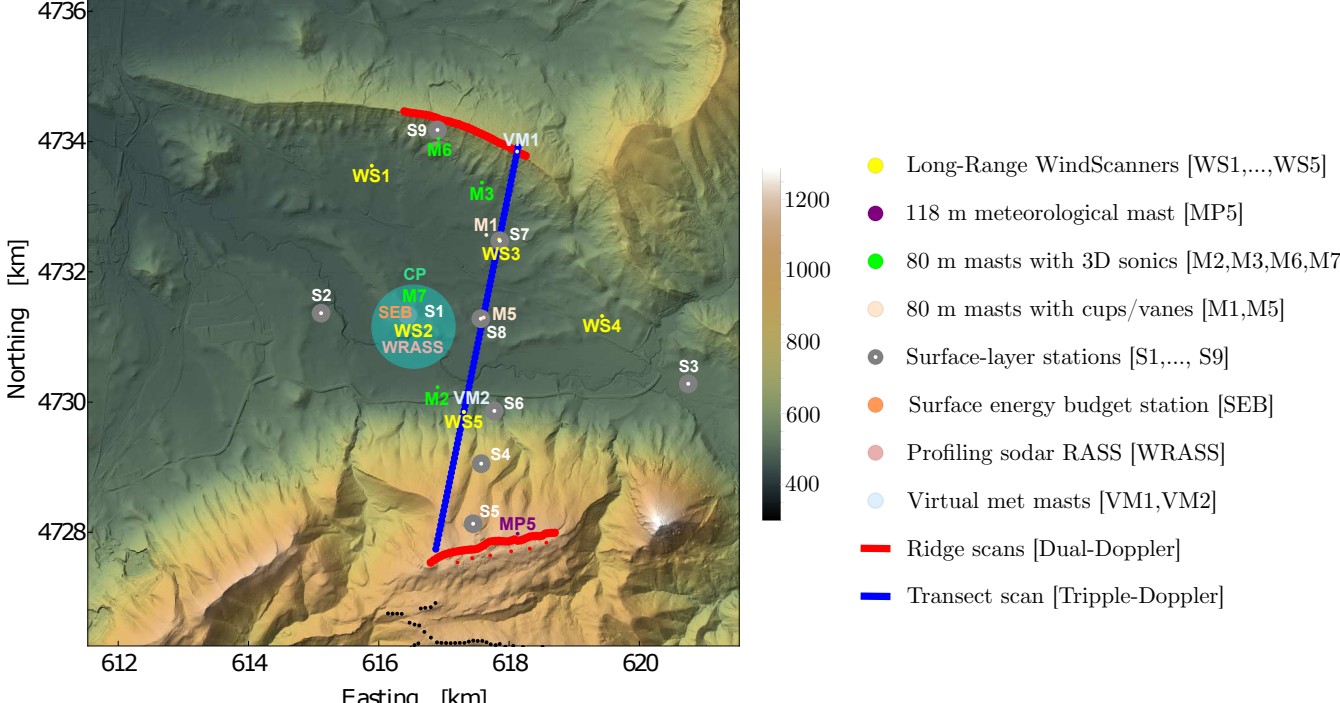

**Figure 3.** Instrumentation layout of the ALEX17 campaign within the 10 x 10 km experimental domain centered at CP. The legend details each type of measurement station. The transect scan (blue line), detailed in figure 4, is the focus of this study.

north ridge slopes. They provide wind speed, wind direction, temperature and turbulent momentum and sensible heat fluxes (for the ones with sonics) profiles at five main levels: 10, 20, 40, 60 and 80 m. In this study, we use 3D sonic anemometers and air temperature measurements from M3, M7 and M2, located at the north ridge slope, CP and Alaiz foothills respectively.

**Surface-layer stations (S).** Due to the heterogeneous land cover, estimation of the horizontal distribution of surface fluxes
5    along the valley and mountain slopes became a need. Nine surface-layer stations, shown with grey markers in figure 3 were deployed for this. Each station had a 2D sonic anemometer at 2 m agl, two levels of temperature (0.36 and 2 m agl) and soil heat flux measurements at a depth of 0.08 m along with soil moisture and temperature at a depth of 0.05 m. These stations were used in a previous study to successfully characterize the spatial variability of atmospheric and soil patterns close to the surface at the hectometer scale (Simó et al., 2019).

10    **Central Position (CP).** This is considered a major location of the experiment due to its density of instruments, see figure 3. The CP is located in the middle of the valley, with relative flat surroundings and farmland as the predominant land cover. Apart from WS2 and M7, there were: i) a radar acoustic sounding system, WindRASS, capable of measuring wind and virtual temperature profiles up to 400 m agl; ii) a surface energy budget (SEB) station, able to estimate the four main components of the surface energy balance (i.e., net radiation, sensible and latent heat fluxes, and ground heat flux). Previous studies in the Ebro





valley used the SEB to quantify significant imbalances in the energy budget (Cuxart et al., 2015) and observed the occurrence of low-level jets and katabatic winds with the WindRASS (Cuxart et al., 2012).

Cantero et al. (2019) documented the ALEX17 campaign, with further and more detailed technical specifications of each of the instruments, together with their geographical coordinates, as well as their operating periods and data availability.

## 2.3 WindScanner measurements

The planning of ALEX17 WindScanners scanning strategies was built upon the experience of previous experiments (Pauscher et al., 2016; Vasiljević et al., 2017; Menke et al., 2019). ALEX17 used a combination of five long-range wind scanners (LRWS), which can be collocated with a pointing accuracy of up to 0.05 deg and synchronized in time within 10 ms (Vasiljević et al., 2016). The campaign lasted for almost 9 months from May 2018 to January 2019. The pointing error during the IOP was kept within 0.2 deg based on regular hard target mapping (c.f. Table 4 in Cantero et al. (2019)).

To execute the ridge, transect and virtual meteorological masts scans shown in figure 3, a total of seven scanning strategies were designed and programmed. With more planned scans than available lidars, each system was scheduled to perform a cycle of three scanning strategies, each lasting 10 minutes, i.e. all trajectories are completed at least twice per hour (four times for the north ridge scan). Table 1 shows the final schedule for each WindScanner.

| WindScanner | 1st 10-min interval | 2nd 10-min interval | 3rd 10-min interval |
|---|---|---|---|
| WS1 | North ridge scan † | Virtual mast (VM1) ‡ | North ridge scan † |
| WS2 | Virtual mast (VM2) ‡ | Transect scan † | South ridge scan † |
| WS3 | North ridge scan † | Transect RHI ‡ | North ridge scan † |
| WS4 | Virtual mast (VM2) ‡ | Transect scan † | South ridge scan † |
| WS5 | Virtual mast (VM2) ‡ | Transect RHI + VM1 ‡ | Transect RHI ‡ |

**Table 1.** Half-hourly repeating schedule of scanning strategies. Scans with † are fully synchronized and those with ‡ start simultaneously at every interval.

The ridge scans, shown as red lines in figure 3, were each composed of 40 evenly distributed points 50-m apart, which followed the terrain profile at 125 m agl. Table 1 shows which pairs of LRWS were programmed to measure synchronously by traversing the beams through the ridgeline points (†), i.e. dual-Doppler measurements. The virtual meteorological masts were defined by the intersection of range-height indicator (RHI) scans. VM1 extended up to 1400 m agl and was measured with WS1 and WS5 during the 2nd interval, whereas VM2 went up to 1200 m agl and consisted of three RHI scans performed by WS2, WS4 and WS5 (tripple-Doppler). The RHIs were coordinated, i.e. not fully synchronized ‡, with the laser beams not visiting the same points of the virtual mast at the exact same time.

Figure 4 shows a cross-section of the transect scan, defined by the vertical plane spanned by WS3 and WS5. The transect scan (blue dots) followed the terrain profile (grey area) at 125-m agl. The scan is measured synchronously by WS2 and WS4 at 85 equally distributed points 50-m apart, starting at the north ridge (location of VM1, light blue markers) and ends at the top



of the Alaiz mountain, where it met the westernmost point of the south ridge scan. Two overlapping RHIs with opening angles of 15° and 20°, respectively, extend the scan curve upward. See table A1 for further details of these RHI scans.

All RHIs had a range of 5 km, but valid measurements are dependent on data filtering criteria that disregard regions e.g. with low clouds and fog, especially near the Alaiz mountain ridge. Measurements from some meteorological masts are here used to
5 explain flow patterns measured by the RHIs. Hence, the positions of M3, M7, M2 and MP5 were projected onto the transect plane as reference (black dots).

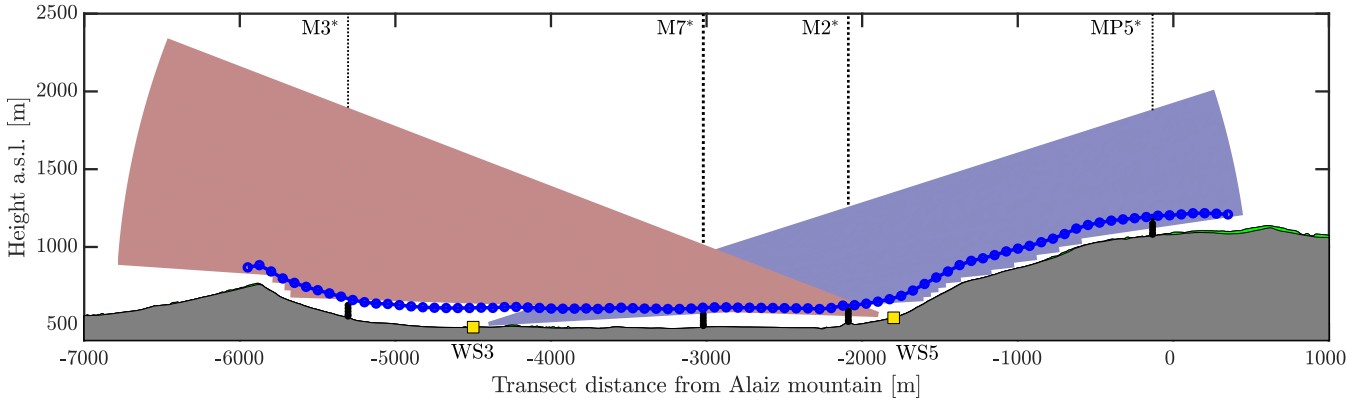

**Figure 4.** Transect scan on the vertical plane spanned by WS3 and WS5 (yellow squares). Shaded areas represent the superimposed RHI scans after a hard target filtering. The blue dots show the transect scan measured by WS2 and WS4 (out of the plane). The black dots indicate the positions of the masts M3, M7, M2 and MP5 when projected onto the transect's plane.

Each RHI scan in figure 4 took ≈30 s, so we ensemble-average 20 scans per 10-min period, computed twice every hour. Before averaging a scan, two noise filters are applied: i) a hard target filter, which finds range gates with carrier-to-noise ratio (CNR) larger or equal to 5 dB and removes all range gates beyond this point (see figure 4) and ii) the variation of the radial
velocity $v_r$ between consecutive range gates along each line-of-sight (LOS) is estimated to filter out values larger than 1.5 ms$^{-1}$.

This multi-lidar scanning patterns provided 2D and 3D wind reconstruction over the ridge scans (red lines) and transect scan (blue line), respectively. The combined Z-shaped transect (figure 3) had a length of 10 km and is a unique feature of this experiment. Additionally, after the IOP all systems performed a one-month campaign staring at M7's 80-m 3D sonic storing
in addition to the wind speeds also the raw Doppler spectra. The Z-shaped transect's UTM coordinates can be found in the Supplement.





## 3 Site wind conditions

### 3.1 Site Climatology

The reference meteorological mast MP5 has eight years of continuous wind speed and wind direction measurements from a calibrated cup anemometer and wind vane at 118 m agl. Figure 5a shows the wind climatology with a wind rose from the

period between February 2011 to January 2019. The Alaiz mountain top presents a mean wind speed of 8.6 m s$^{-1}$ at 118 m with a turbulence intensity of 7% at 15 m s$^{-1}$ as well as a bidirectional regime, with prevailing winds from a north-northwest sector (330°–360°, 32% of total) and a south-southeast sector (150°–180°, 23% of total).

Figure 5b shows the wind rose of ALEX17 two years' extensive measurement period, from July 2017 to July 2019, which has a representative wind regime when compared with the long-term climatology. As the MP5 is located next to the wind

turbines during power curve measurements, it is susceptible to wind turbine wake effects with winds between 130° to 240° (see figure 3).

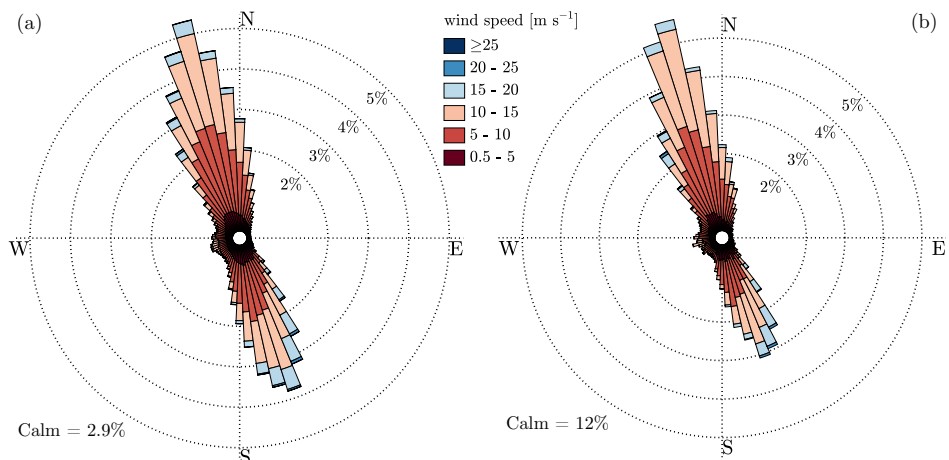

**Figure 5.** Wind roses of 10-min wind speeds observed by MP5 at 118 m for the reference period (2011-2019) **(a)** and during the EMP **(b)**. The calm threshold is 0.5 m s$^{-1}$.

In complex terrain, the characterization of spatial variability is critical to capture a full picture of the wind regime. Figure 6a illustrates this with the wind rose measured at the valley floor by M5 during the EMP. Comparing the mountain top (Fig. 5) with the valley floor (Fig. 6a), we observe a turning effect with NW and SE valley winds. Higher wind speeds come from SE

confirming the funnel effect caused by the valley conical section with a larger area facing west (see figure 2b).

The combination of topographic features $\mathcal{O}(10$ km$)$ away and elevation changes up to 700 m between the valley floor and mountain top poses a challenge to numerical models such as the Weather Research and Forecast (WRF) model. The NEWA project delivered a wind atlas for all European countries using WRF with a horizontal and temporal resolution of 3 km and 30 min, respectively (Hahmann et al., 2020; Dörenkämper et al., 2020). Here called NEWA-WRF, this model output is publicly

available and covers a 30-yr period (1989–2018).





Figure 6b compares the wind distribution measured at MP5 during both the long-term reference period (February 2011–January 2019) and the EMP (July 2017–July 2019) with the simulated distribution from NEWA-WRF also from the long-term period. The model output was extracted at the MP5 position using a linear interpolation of the nearest neighbor grid cells as well as the nearest vertical levels. Results show that the NEWA-WRF simulations underestimate the mean wind by more than
$1.5~\text{ms}^{-1}$ which is indicative of unresolved speed-up effects in the meso-scale model. The measured wind distribution further confirms that the EMP can represent the wind climate.

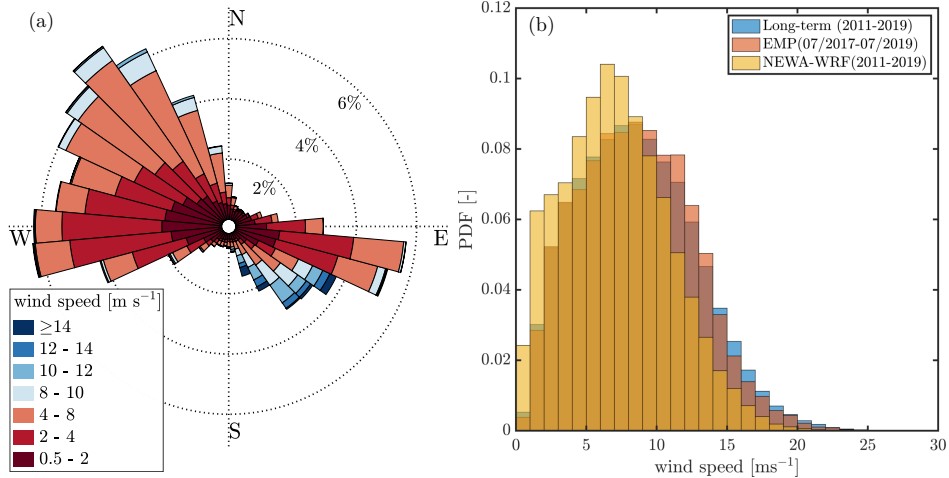

**Figure 6.** Panel **(a)** shows the wind rose of 10-min wind speeds at the valley measured by M5 at 80 m agl. Panel **(b)** shows the wind speed distributions at 118 m agl measured at MP5 during the reference period and EMP along with the modeled wind speeds from NEWA-WRF for the reference period.

An extensive study evaluated the mean wind speed bias between measurements and NEWA-WRF simulations using 291 onshore masts over Europe, where complex sites are defined as having 2% of their slopes higher than $16.7°$ within a radius of 3.5 km (Dörenkämper et al., 2020). Results showed that the selected NEWA-WRF setup in complex sites has a mean wind
speed bias of $-0.25 \pm 0.83~\text{ms}^{-1}$. Based on Fig. 6b, ALEX17 has a much larger systematic underestimation of the wind speed when compared to NEWA-WRF's validation study, as expected since the complexity at MP5 is ten times higher based on the same criteria. In order to improve modeling here we need to further downscale NEWA-WRF using non-linear micro-scale models.

### 3.2 Atmospheric stability

Monin-Obukhov Similarity Theory (MOST) is commonly used to describe the mean and turbulence characteristics of the flow within the surface layer (Foken, 2006). MOST assumes that over horizontally homogeneous and flat terrain (HHF) conditions, normalized atmospheric gradients such as those of wind and temperature are functions of the dimensionless parameter $z/L$, where $z$ is the height above ground and $L$ is the Obukhov length, which is defined as $L = -(u_*^3 T_s)/(\kappa g \overline{w'T'_s})$, where $u_* =$





$(\overline{u'w'}^2 + \overline{v'w'}^2)^{1/4}$ is the friction velocity, $\kappa = 0.4$ the von Kármán constant, $g$ the acceleration due to gravity and $T_s$ the sonic anemometer temperature. The prime ($'$) denotes fluctuations and the overbar a mean. The velocity components $u$, $v$ and $w$ refer to the stream-wise, cross-wind and vertical velocity components, respectively.

In ALEX17 we select the central position as a reference for atmospheric stability and we use $z/L$ to evaluate the wind profiles under different stability conditions for two predominant wind directions. The CP is located on a plateau within the valley floor surrounded by farmland, which is one of the flattest and most homogeneous areas in the experimental area, but still far from HHF conditions. The climatology of stability is derived by computing $z/L$ at 10 m agl from sonic anemometer measurements at 20 Hz, computing the fluxes over 30 minutes. We use the coordinate system described above by applying yaw and pitch rotations, i.e. double rotation. Table 2 shows the atmospheric stability classes given by $z/L$ intervals following the analysis by Berg et al. (2011) and the number of 30-min profiles.

| Stability class | $z/L$ | # of 30-min profiles |
| --- | :---: | :---: |
| unstable (u) | $z/L \leq -0.2$ | 572 |
| near unstable (nu) | $-0.2 < z/L < -0.04$ | 904 |
| neutral (n) | $|z/L| \leq 0.04$ | 1447 |
| near stable (ns) | $0.04 < z/L < 0.2$ | 1115 |
| stable (s) | $z/L \geq 0.2$ | 1054 |

**Table 2.** Definition of atmospheric stability classes within intervals of $z/L$ and respective quantity of 30-min profiles.

The M7's 3D sonic anemometer closest to the surface is at 10 m agl. One-year of measurements (from August 2018 to July 2019) are available from this sonic anemometer with $84.6\%$ of valid 30-min periods. Figure 7a shows the daily cycle of dimensionless stability $10/L$ at M7. As expected, stably stratified conditions prevail at night, whereas unstable conditions dominate during daytime and peak around midday. Figure 7b shows the behavior of stability with wind speed, where a prevalence of neutral conditions with increasing wind speeds is found.

Figure 8 presents the vertical profiles of the normalized mean wind speed for two prevalent wind sectors at the valley floor, namely a NW sector (300°±30°) and a SE sector (120°±30°), both divided by stability classes. For the SE sector (Fig. 8a) large differences can be observed close to the surface, showing that the surface roughness' fetch can be quite inhomogeneous for this sector but also likely influenced by seasonal effects. Also, negative wind shears characterize the stable class, which are potentially caused by valley drainage flow (Serafin et al., 2018). Profiles from the NW wind sector (Fig. 8b) resemble more flat and homogeneous conditions, with increasing wind shear with stability and a similar normalized wind close to the surface except for stable conditions.

Furthermore, when dealing with mountain flows, the topography occupies a large portion of the ABL and hence plays a major role in the flow stratification. Disturbances in the stable atmosphere caused by the topography may generate three-dimensional flow phenomena i.a. atmospheric lee waves, rotors and hydraulic jumps (Kaimal and Finnigan, 1994). The natural frequency of these vertical oscillations is characterized by the Brunt–Väisälä frequency $N = [(g/\overline{\theta})(\partial\overline{\theta}/\partial z)]^{1/2}$ where $\partial\overline{\theta}/\partial z$



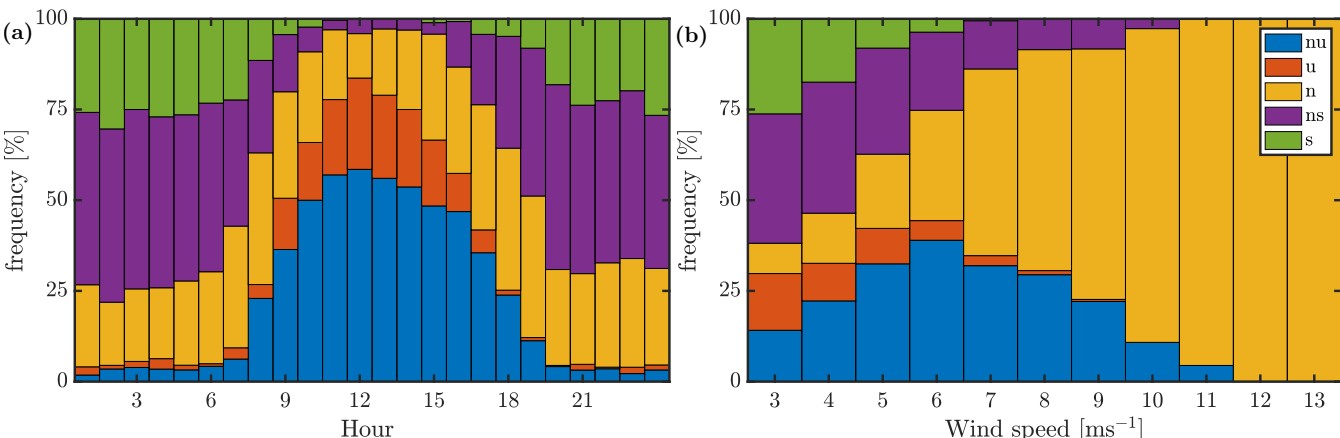

**Figure 7.** Frequency of $z/L$ at 10 m agl divided in stability classes per hour **(a)** and per wind speed **(b)** from measurements at M7. The stability classes are found in table 2.

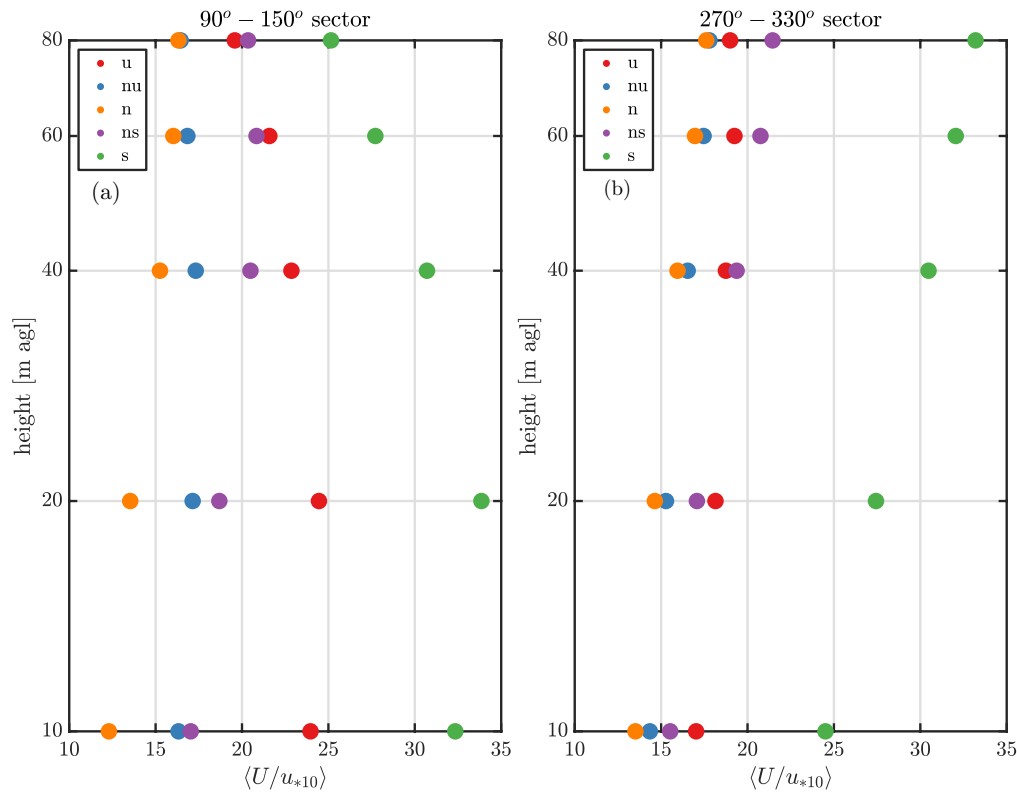

**Figure 8.** Mean vertical wind profiles normalized with the friction velocity computed by the 10-min sonic anemometer ($u_{*10}$). Panel (a) shows profiles within the $120°\pm30°$ sector and panel (b) shows profiles within the $300°\pm30°$ sector based on wind directions at 10m.





is the potential temperature gradient. The potential temperature is computed as $\theta = T + 0.0098z_{asl}$, where $0.0098 \text{ Km}^{-1}$ is the dry adiabatic temperature gradient.

The Froude number (Fr) is the dimensionless parameter given by the ratio of flow inertia based on a reference upwind wind speed $U$ to the gravity forces acting on the flow (Kaimal and Finnigan, 1994). Following the linearized solution presented by
Rotunno and Lehner (2016) $\text{Fr} = \pi U/2ND$, where D is a height for the stably stratified flow layer upstream of the mountain, also called inversion depth. The scaling can also be performed with a characteristic mountain height H.

We select the characteristic height H as the elevation difference between MP5 and M7, which is $\approx 500$ m. The inversion depth D is assumed to be proportional to the high wind speed layer height from the lidar measurements (c.f. figure 9), hence $H \sim D \approx 500$ m. Fr is evaluated at the Alaiz mountain top (MP5) and foothills (M2), at the valley floor (M5) and at the north
ridge slope (M3) using the same $D$. The potential temperature gradient in $N$ is calculated by linear interpolation between the measurements at 2 and 80 m for M5/M3 and between 2 and 113 m for MP5. Here our aim is to investigate the relative changes in Fr, where we compare Fr estimations at the mountain top and along the valley.

## 4    Selected flow cases

### 4.1    The Lee-Side Hydraulic Jump

An atmospheric hydraulic jump occurs when a two-layer stratified flow encounters an obstacle and experiences a discontinuity in the flow layer depth and velocity, causing dissipation of kinetic energy in order to recover part of the original potential energy that existed upstream. It differs from atmospheric gravity waves or lee waves since it involves a discontinuity and requires non-linear dynamics to be described, analogous to a shock wave when sound crosses the Mach number. Hence, it is assumed that the hydraulic theory can describe these high-amplitude mountain waves with downslope flow in the lee side (Baines, 1995).

Long (1954) performed water tunnel experiments and his results show that the lee-side hydraulic jump occurs when the topographical feature is comparable to the depth of the upstream flow layer and there is a Fr transition between supercritical ($> 1$) at the mountain top and subcritical ($< 1$) in the lee-side. The solutions presented by Houghton and Kasahara (1968) and Vosper (2004) also predicted under which conditions the jump occurs, based on $H$, $D$ and a Fr scaled with the maximum wave speed given by $\sqrt{gD}$.

METCRAX II identified jump-like episodes using co-planar RHI lidars scans (Lehner et al., 2016). Whiteman et al. (2018) used the latter lidar measurements to propose a conceptual model of the jump, yet limited to the scales and particularities of the meteorite crater case study. Rotunno and Bryan (2018) further performed numerical simulations of a steady-state jump considering the full time-dependent three-dimensional flow description. Results quantify the jump's evolution and structure based on potential temperature, vorticity and turbulent kinetic energy.

We defined $H \sim D \approx 500$ m in section 3.2, which agrees with previous studies (Sanz Rodrigo et al., 2013). Figure 9 shows four 10-min periods of co-planar RHIs during one night where a hydraulic jump was spotted at the lee side of the Alaiz mountain between October 5[th] and 6[th] 2018.

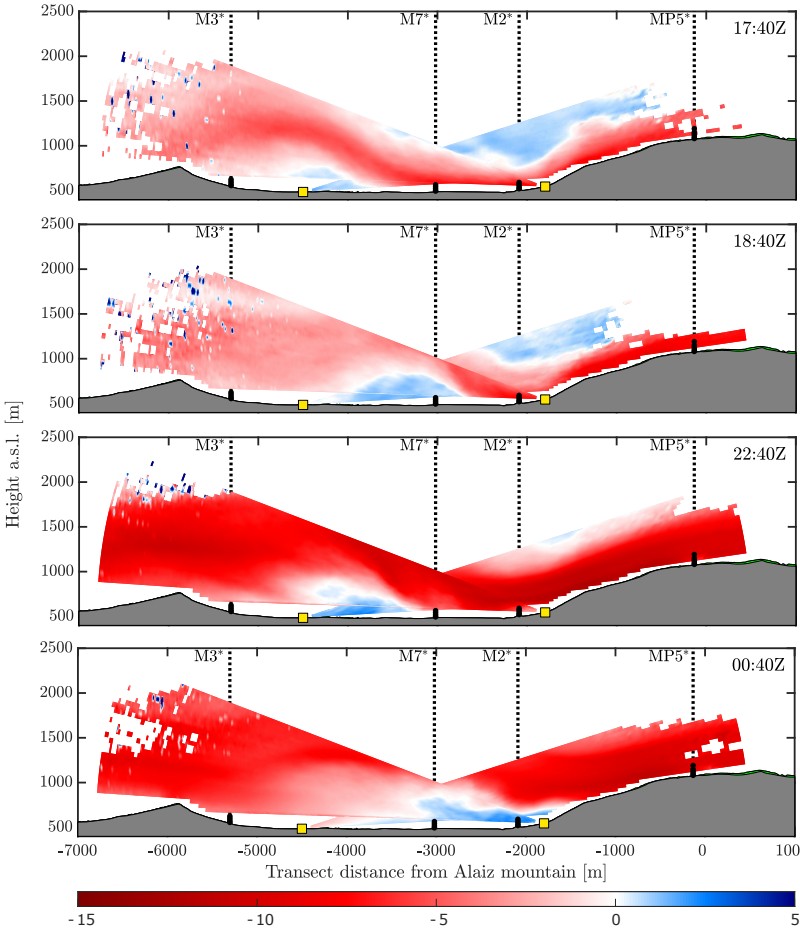

**Figure 9.** Superposition of dual RHI scans during the hydraulic jump period between 5$^{th}$–6$^{th}$ October 2018. The color bar represents radial wind speed in ms$^{-1}$ with negative values representing flow from right to left. Each of the frames correspond to a 10-min scan.

The main southerly flow has negative radial wind speeds, indicated by red colors, where a strong downslope wind reaches the valley and performs the jump, which is non-steady as the jump location changes with time. Long (1954) also argued that when Fr decreases at the obstacle, the jump moves towards the obstacle (mountain) and loses intensity. Conversely, when Fr increases at the obstacle, the jump moves further downstream and becomes more intense.

Figure 10 compares the wind direction regimes and Fr numbers between the masts located at the mountain top (MP5), mountain foothills (M2), valley floor (M5) and north ridge's slope (M3). The M7 mast did not have valid temperature measurements during this period, hence it is not shown. The southerly flow is maintained at MP5 throughout the period, with easterly winds at the jump's location. We observe a transition between Fr> 1 at the mountain top to Fr< 1 at the valley. According to Long (1954), the supercritical to subcritical transition is strong evidence for the onset of a hydraulic jump, which is confirmed herein as well as in smaller scale experiments (Rotunno and Lehner, 2016).





Furthermore, the time evolution of the hydraulic jump described by Long (1954) is also confirmed in this episode. Between 18:40Z and 22:40Z the jump tends to move further downstream as Fr at MP5 goes from 1 to 1.5. From 22:40Z to 00:40Z the flow loses intensity, the jump moves towards the mountain while Fr at MP5 decreases from 1.5 to 1.2, and the M2 measurements evidence a recirculation zone in the lee-side of the mountain.

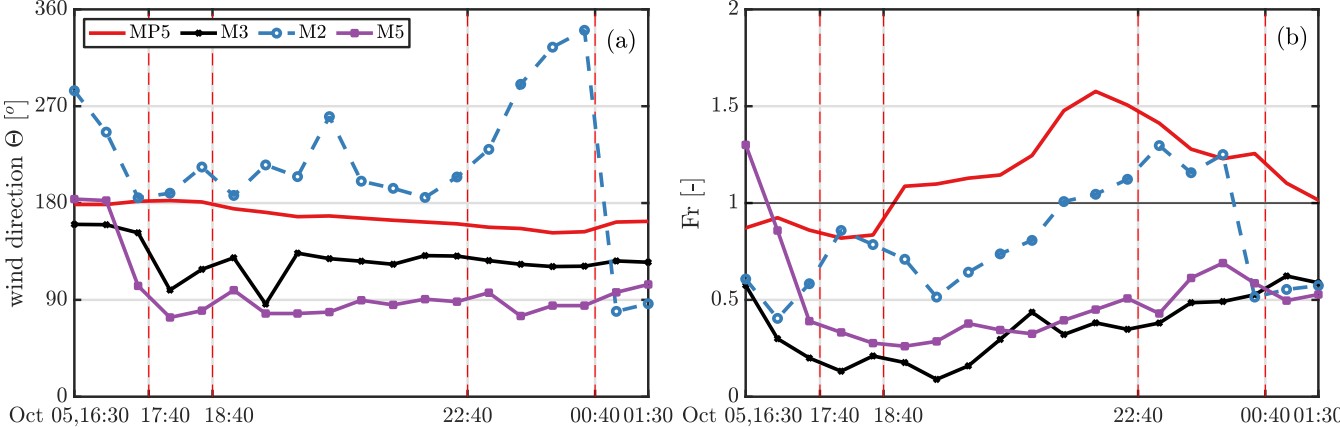

**Figure 10.** Time series of wind direction **(a)** and Froude number **(b)** during the hydraulic jump case. The vertical dashed lines represent the snapshots of figure 9.

Figure 11a shows a 10-min average snapshot (19:00Z) of the hydraulic jump episode, with wind and temperature measurements from the surface-layer stations. The stratified southerly winds close to the surface enter the Elorz valley surrounding the Alaiz mountain through the eastern side. The flow at the center of the valley (from S3, S8, S1 and S2) is from the east. The wind speed from the along-valley stations also shows the effect of topographical channeling, i.e. the wind decelerates as the valley becomes wider. On the other hand, the flow along the Alaiz slopes comes from the south (see S5, S4 and S6) and remains

almost steady along the night. Wind at the northern part of the valley is weak, which is also observed by the lidar scans, and the air stratification is more intense, as shown by S7 and S9, located downstream the hydraulic jump.

     Figure 11b shows the same 10-min snapshot (19:00Z) of wind speed and direction profiles from S1, M7, M5 and sodar RASS at the central position. The instruments at the CP agree and show quiescent easterly winds. Additionally, the wind speed profile at M5 further highlights the valley channeling effect, with measurements located 1 km east and 20 m below CP. The

potential temperature profiles (figure 11c) from S1, M5 and sodar RASS agree and show a stable boundary layer. The presence of a stratified valley floor with stagnant flow agrees with similar observations of METCRAX II (Lehner et al., 2016), but here in a scale ten times larger.

     This case shows that the lee-side hydraulic jump is characterised by a Fr transition from supercritical Fr> 1 at the mountain to subcritical Fr< 1 at the valley floor. After inspecting the entire year period with concurrent data from MP5 (mountain top)

and M5 (valley floor), approximately 35% of all the stably stratified periods reproduce a similar Fr transition. Thus, considering





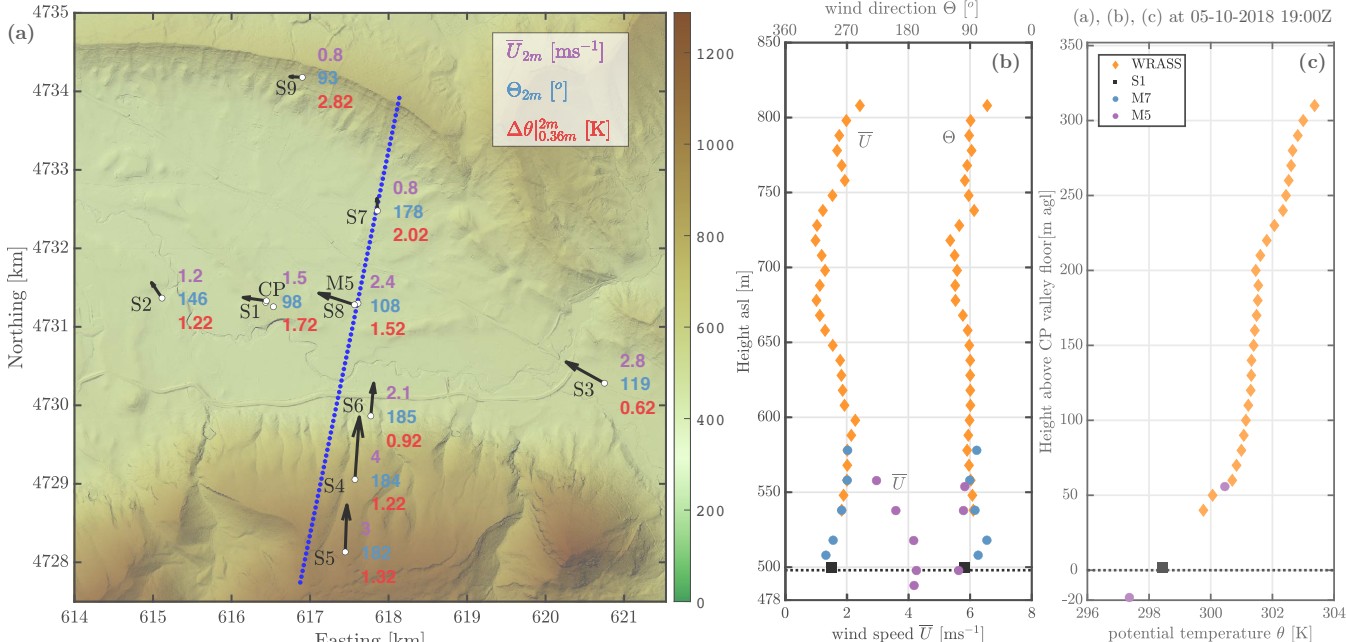

**Figure 11.** Wind conditions during hydraulic jump case at 05-10-2018 19:00Z. Panel **(a)** shows surface-layer stations' wind and potential temperature measurements. Panels **(b)** and **(c)** show vertical profiles of wind and potential temperature at the valley from the sodar RASS, M7, S1 and M5. The dashed line marks the valley floor level at CP.

only southerly winds, the occurrence of the hydraulic jump could reach 10% of the time, suggesting that the current dataset could be suitable for a deeper study on such phenomena.

## 4.2 Layered flow induced by valley winds

During the hydraulic jump we have seen how winds at the valley characterized the jump's recirculation zone. Disparately, with
5 low wind speeds thermal stratification can modulate flow patterns (Jiménez et al., 2019), especially due to the heterogeneous land cover, which causes unequal heat fluxes (Martínez et al., 2010). Grubišić et al. (2008) showed a three-layer flow structure in a valley-mountain region under quiescent conditions, where up-valley and down-valley winds are thermally driven and decoupled from the meso-scale flow aloft. ALEX17 also presents valley winds that develop into layered stratified flows.

An example is found during the night of October 24$^{th}$ to 25$^{th}$ 2018. For this case, the synoptic situation is dominated by
10 a strong wind of northern component whose intensity starts to decrease several hours after sunset. Closer to the surface, this general wind takes a western direction within the Elorz valley. Figure 12a shows four 10-min periods of the co-planar RHI scans representative of the evolution of the layered flows within the valley. As in figure 9, the positive radial wind speeds represent flow from left to right. Figure 12b displays the corresponding potential temperature profiles measured by the sodar





RASS and M5 at the valley center, where the air becomes cooler close to the surface and a thermal inversion develops along the night, becoming very intense after midnight.

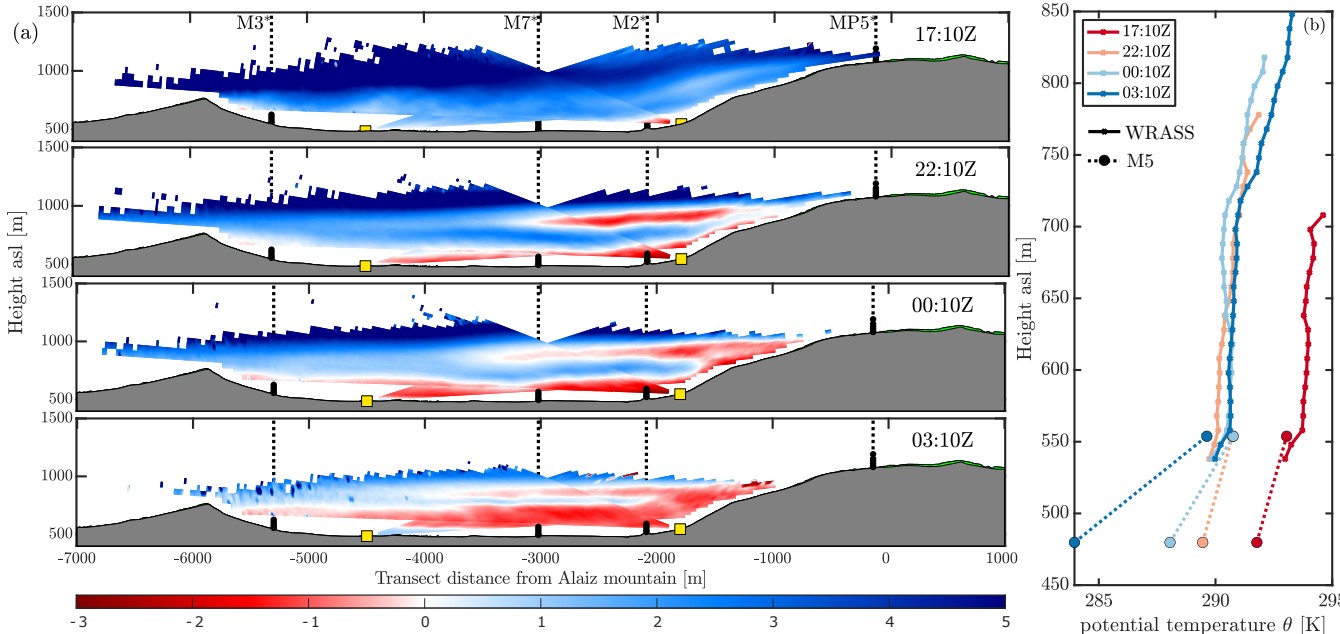

**Figure 12.** Panel **(a)** shows the superposition of dual RHI scans during the layered valley winds period between 24th–25th October 2018. Panel **(b)** shows the potential temperature profiles measured by the sodar RASS and M5 at each frame in **(a)**.

Before sunset (≈17Z), the northwesterly flow within the valley is coupled with stronger mountain winds aloft as shown by the first lidar scan. A weak stably stratified layer develops after sunset and surface winds adjust to a new equilibrium where the valley topography has a major influence. This layer depth increases as the thermal inversion develops within the valley, producing a layering effect as seen by the subsequent RHI scans (figure 12a). Southwesterly winds are present at the valley floor and over the southern slopes, while a northwesterly flow remains within the valley atmosphere aloft, still coupled with the mesoscale wind. Both layers are stably stratified although with different intensity. The thermal stratification at the upper layer remains steady along the night, while the surface inversion evolves very slowly until midnight (figure 12b).

Figure 13a shows a 10-min average snapshot (22:30Z) of the valley winds measured by the surface-layer stations, indicating the southerly component of the wind at the valley center and over the southern slope within the surface layer, as well as the increment of the wind speed as the valley gets narrower towards the east. The S7 and S9 positions show a recirculation zone in the lee-side of the north ridge, where the wind is very weak together with the strongest surface thermal inversion.

Figures 13b and c represent the wind speed and wind direction measured at MP5 118 m agl and at 80 m agl by distinct masts across the valley. There is a persistent offset of 90° in the wind direction between the mountain (MP5) and valley (M2,M3,M5), while the general wind decreases throughout the evening, as indicated by MP5. In consequence, the wind speed also diminishes within the valley, intensifying the surface cooling and generating a stronger thermal inversion at the valley floor after midnight

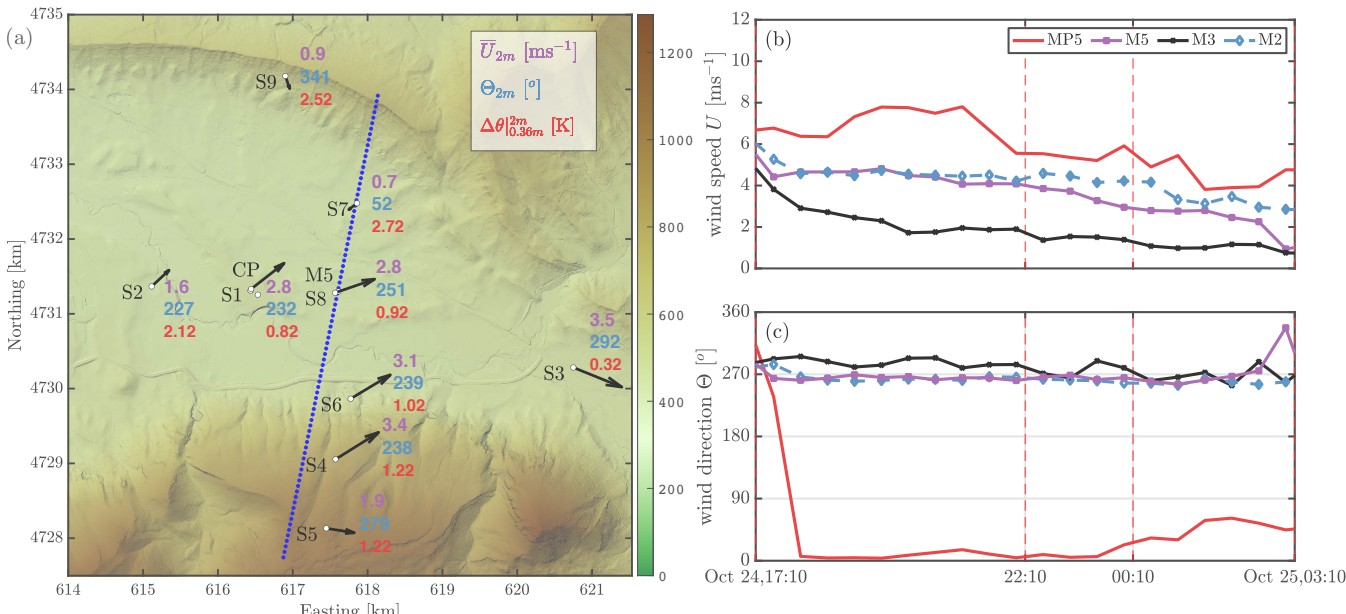

**Figure 13.** Panel **(a)** shows surface layer stations' wind and potential temperature measurements at 24-10-2018 22:30Z. Panels **(b)** and **(c)** show the time series of wind speed and direction from the masts, respectively. The vertical dashed lines represent the snapshots of figure 12.

(figure 12b). This situation decouples the surface layer, with winds responding to a local regime, from the southwesterly flow within the valley (not shown). An elevated thermal inversion around 720 m asl decouples the valley atmosphere from the northerly wind aloft (see profile at 03:10Z in figure 12b), increasing the depth of the red band within the valley observed in the RHI scan at 03:10Z.

## 5   Summary and Conclusions

This work presents the Alaiz Experiment (ALEX17) with details on the main innovations in atmospheric measurements, unique characteristics and two flow patterns observed during the eight-month Intensive Observational Period (IOP). ALEX17 is in a striking position in terms of scale and complexity. The experimental domain is large enough for synoptic effects to be essential for flow modeling while still being within the range of current long-range commercial lidars.

Additionally, ALEX17 requires the adoption of meso- to micro-scale models. Palma et al. (2019) showed advancements in such modeling schemes during the Perdigão experiment, especially for stably stratified flows. ALEX17 yet poses a further challenge in numerical modeling efforts for its large domain size.

The topographical features of ALEX17, with a non-parallel mountain-valley-ridge configuration, lead to the occurrence of valley winds with quite distinct wind regimes from the mountain top, where wind direction offsets up to 90° and channeling effects being observed. To capture most of the wind flow spatial variability, five WindScanners performed synchronized mea-





surements of a unique 10-km transect line, covering a vast portion of the experimental domain. The experimental report offers further technical information on the campaign and dataset (Cantero et al., 2019).

Wind measurements at the mountain top are available since 2011, nevertheless ALEX17's two years extensive measurement campaign has been able to represent well the wind climatology. When comparing available meso-scale modeling from the New
European Wind Atlas with observations, a negative bias in the mean wind larger than 1.5 m s$^{-1}$ pointed to the need of further improvement of model performance at this site. At the valley floor, vertical wind profiles with southeasterly winds showed negative wind shear under stable conditions, whereas for northwesterly wind profiles are more canonical but still affected by land cover inhomogeneities.

Using the multi-lidar scans we spotted a lee-side atmospheric hydraulic jump episode. Measurements from masts and
surface-layer stations corroborated the formation and evolution of the jump with long-established predictions (Long, 1954). The Froude number transition from supercritical ($> 1$) to subcritical ($< 1$) was quantified and results suggest that this type of stratified atmospheric wave can potentially happen during $\approx 10\%$ of the time in this site.

The hydraulic jump is connected with high southerly winds at the mountain top. On the other hand, during stratified conditions with lower wind speeds, valley winds become decoupled from the mountain flow aloft due to thermal stratification. This
case is illustrated with a layered flow pattern measured by the lidar scans, where westerly stratified valley winds interact with northerly winds aloft. Other atmospheric phenomena captured within the IOP shall be the object of further studies.

We have characterized two multi-scale stratified flows regarded as challenging both in terms of its physical description (Serafin et al., 2018) and modeling (Sanz Rodrigo et al., 2017). Other flow episodes, featuring i.a. upstream flow blockage and lee waves, were spotted and can be the object of further studies. These measurements will provide aid to evaluate numerical
models especially for testing the effects of atmospheric stability on flow over complex terrain.

*Data availability.* The Alaiz Experiment dataset collected during the EMP is publicly available and can be found in Santos et al. (2019), with metadata and citation guidelines for each group of instruments.

## Appendix A: Lidar configuration for RHI transect scan

*Author contributions.* Funding acquisition and project administration: J.M., J.S.R., J.C., E.C. Conceptualization and resources: J.M., J.S.R.,
E.C., P.S., N.V., J.C. and D.M-V. Data curation: P.S., D.M-V., F.B.G. and E.C. Methodology: P.S., N.V. and D.M-V. Formal analysis and visualization: P.S. and B.M. Writing - original draft: P.S. Writing - review & editing: P.S., J.M., N.V., E.C., J.S.R., F.B.G., D.M-V., B.M. and J.C.

*Competing interests.* The authors declare no conflict of interest.



| Parameters | WS3 | WS5 |
|---|---|---|
| (Easting, Northing) [m] | (617846.80,4732496.43) m | (617305.59,4729848.75) m |
| Elevation [m asl] | 487.21 m | 545.92 m |
| Azimuth [$^o$] | 191.55° | 11.55° |
| Elevation (min,max)$^o$ | (3°,18°) | (1.58°,21.56°) |
| Scan speed [$^o s^{-1}$] | 0.5°s$^{-1}$ | 0.666°s$^{-1}$ |
| Range gates [min:$\Delta$:max] m | [100:50:5000] m | [100:20:5000] m |
| Accumulation time [ms] | 1000 ms | |
| Pulse length [ns] | 400 ns | |
| Points per range gate [-] | 128 | |
| Signal spectral width [ms$^{-1}$] | 3 m s$^{-1}$ | |
| Specified physical resolution [m] | 75 m | |

**Table A1.** Windscanner's position and configuration for the Transect RHI scans.

*Acknowledgements.* We would like to thank Michael Courtney and all staff from DTU Wind Energy (Denmark), CENER and UIB (Spain) involved on the commissioning and ongoing monitoring of the experimental infrastructure. We credit Alfredo Peña for the analysis that resulted in Figure 1. The European Commission (EC) partly funded NEWA project (NEWA- New European Wind Atlas, FP7-ENERGY.2013.10.1.2, European Commission's grant agreement number 618122). National agencies that also provided funding are the *Danish Energy Agency*

5    and *Ministerio de Economia y Competitividad* (Spain). B.M. was supported by the grant of the *Vicepresidència i Conselleria d'Innovació, Recerca i Turisme del Govern de les Illes Balears*.





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
