# Peer review of "The Alaiz Experiment: untangling multi-scale stratified flows over complex terrain"

_Wind Energy Science, 2020_

## Referee Comment (RC1) · Anonymous Referee #1 · 23 Aug 2020

General comments:

I enjoyed reading the manuscript. It brings up very interesting flow situations from a modern measurement campaign. It is my impression that the scientists behind the study made a great effort compiling data from, and running, such an extensive campaign. In my judgement the study would be improved by clarifying the aim of the study; Is it to present the campaign? Is it to present important flow phenomena, and in that case, why are they important, and why were these two cases selected? Based on the aim of the study, the background could be improved by focusing on the previous research related to the aim. The basis of distinguishing the hydraulic jump from a lee wave could be strengthened, especially with the RASS showing semi-layered structure in the potential temperature. A linear stability analysis could potentially be useful. I

think the climatological consequences of the findings could be highlighted more. The assessment of the prevalence of conditions linked to hydraulic jump was interesting, and it would be valuable with something similar also for the layered flow.

Specific comments:

Abstract L. 2. Although not necessary, temporal information is also interesting.

Abstract l.4, perhaps path is better. The transect line over the valley is shorter than 10, so I guess you are referring to the whole path, which is a bit misrepresented with "line".

Abstract l.4 For which of the cases, both? That is not clear at this point.

Abstract l.5. Here the reader does not yet know that you are referring to one of your two cases and that the reference to other cases is interesting in terms of similarity.

Abstract. L.9 Perhaps add something on why you think it is well captured by the lidars. So far the reader has only been introduced to the lidar measurements (no other measurements), which means that the flow disconnect could potentially be a measurement problem.

Introduction L1. Does this refer to climatological or time series? For climatological, what is the basis for saying 4 % is low? I understand it may be low compared to historical perspective, but in terms of economy it is perhaps not such a low uncertainty.

P1. L.15. Additional suitable references for this claim. For example Ayotte 2008 for micro scale models, recent work on validation for the NEWA model (perhaps Drenkämper 2020). In terms of analytical work there are also investigations into this by Finnigan and Belcher (2004), and perhaps also the original theory for flow over hill by Hunt.

P1. L.20 As far as I remember the cited literature does not give a summary of the available validation data. I do not so much argue with the claim, but I'm not convinced the cited literature really backs up the claim. In fact, due to the NEWA project, there are quite a few data-sets being produced, so perhaps new data-sets are not so rare

anymore. I do think the cited literature provides a good reference for the need of validation data.

P2. L.16. Please be specific when referring to new insights.

P2. L34 For sure, with such a steep a profile, a linear model is bound be somewhat inaccurate. However, as far as I remember, the performance of the non-linear models was not altogether convincing, which might also be worth mentioning.

P2. L34. As far as I recall, the Bechmann et al (2011) only used neutral conditions. Are you referring to Berg et al. (2011)? In that case variation in flow pattern or something similar might be better, since I don't think there were any flow model validation in that study, and hence errors might be misleading.

P4. L4. Is there a reference for the airborne laser scans?

P.5 L.16. I don't think only WLS70 is enough information to characterize the lidar. Consider also including the brand name.

P5 L.24 I suppose the availability of the WLS70 is also interesting if it is reported for the tower. Tower width and boom lengths are good to report in my view, as they provide good indication on flow distortion magnitude. Also, data acquisition rates and averaging periods are good to report here if they are not reported further down the text.

P6. L2. What type of sonic? And also same as above. Boom length and tower widths are potentially interesting.

P6. L.13. Reference for the RASS system is missing.

P7. L3. Consider moving this reference to the beginning of the instrumentation section.

P7 L10. Consider also mentioning what maximum dz difference this implies. On the other hand, I suppose also errors in the azimuth angle are interesting.

P8 L10. From this I take it you first average the radial wind speeds and then compute

horizontal vectors? This could be clarified in the beginning of the paragraph, as I suppose the option of calculating horizontal wind vectors and then averaging also exist. The earlier mentioning synchronization makes it a bit unclear.

P10 L6. "Can represent the wind climate" is perhaps a bit general. I see what the authors mean, but on the other hand even 8 years leave room for some uncertainty regarding the true wind climate (in terms of for example the true long term average wind speed). In terms of containing a representative set of meteorological conditions I agree with the authors.

P10 l.14 As in 20 % of the slopes higher than 16.7 %?

P10, final line: Which is defined here as...

P11 L11. The reference height of 10 m is rather low for the scales of this paper, I suppose? The scales in Berg et al (2011) were much smaller. This means the stratification at levels relevant to wind turbines will generally be much stronger, since those heights are a magnitude larger. I think it is useful also to use 10 m, but some discussion might be warranted. Did the authors try to use Ri number from the RASS to assess large scale stratification?

P11 L15. Are the wind speeds also at 10 m height? The same point as above could be made here as well. One could argue that for wind turbine applications the dependence of wind speed with stratification should be representative for turbine heights. Fig. 7 may give the impression that for higher wind speeds it is almost exclusively neutral conditions, while for wind turbine relevant heights the situation might be different.

Figure 7. Since the experiment was in Spain, it might be good to mention what time is referred to for the x-axis. Local (sun) time, local (Spain) time, or UTC? Why does NU come before U?

Figure 8. In my opinion the figure would be improved by adding confidence bounds and possibly lines connecting the dots. It would also be valuable to know the filtering

criteria for the averaging, or at least which period that is used.

P.11 L26 Move this to P13 l 10, or move what is there up here (what you use as reference temperature).

P13. L15. Reference for this statement? I rather thought of a hydraulic jump as a continuity phenomenon (including a transition to turbulence, linked to the lower velocity), with subsequent increase of potential energy. But I'm not that familiar with theory. I guess I object to that the atmosphere "wants" to recover potential energy.

Figure 9, and accompanying discussion. Do you have any criteria for distinguishing a hydraulic jump from a regular lee wave? The finite length of the low velocity region behind the jump in Fig. 9 (particularly in panel 2) makes it look to me also a bit like a wave (also the shape of the jet in panel 1).

Fig. 10. Confidence bounds (even if estimated) would help the argument made in the text (that the Froude number really decreases at MP5).

P15. L18. Horizontal scale?

Fig. 12. Did you identify the cause of the layers with reverse (negative) wind speed? It is interesting to me that they are on the downstream mountain side. Is it drainage currents from the mountain side?

P18 L5. I think in order to say that this study presents the Alaiz experiment, there should be more substance regarding the experimental details. Perhaps introduces is a better word. Also, I think focusing on the flow patterns is interesting enough.

Technical comments The tempus change throughout the text.

---

## Short Comment (SC1) · 31 Aug 2020

In their introduction, the authors provide a nice overview of recent campaigns focusing on flow features in complex terrain categorized after domain size and slope gradient. However, we think that they missed several recent relevant measurement campaigns siting in the center part of their diagram in Fig. 1. We encourage the authors to consider including these campaigns to complement their overview.

One is the PIANO campaign which took place in the Inn Valley over Innsbruck in Austria in 2017 focusing on cold pool and foehn interactions (Haid et al., 2020). Several scanning Doppler lidars were deployed in a complex scanning configuration consisting of coplanar RHI and PPI scans as well as vertical stare mode to get insight into the

mean and turbulent structure of the three-dimensional flow.

Recent coplanar Doppler lidar measurements were conducted in the complex topography around the city of Stuttgart in south-western Germany focusing on mesoscale flow structures and convective cells (Adler et al. 2020).

In 2019, a set of six Doppler lidars was deployed about 20 km east of Innsbruck in the Inn Valley to study the three-dimensional flow structure in the valley under different synoptic conditions. Three of the lidars performed synchronized coplanar RHI scans to capture the cross-valley kinematic flow structure across the whole valley. A BAMS paper about the campaign and measurements is currently under review (Adler et al. 2020). Some information on the campaign were presented during the 19th Conference on Mountain Meteorology (https://ams.confex.com/ams/19Mountain/meetingapp.cgi/Paper/376136) and on the KIT website (https://www.imk-tro.kit.edu/english/844_8306.php).

References:

Haid M., A. Gohm, L. Umek, H.C. Ward, T. Muschinski, L. Lehner, M.W. Rotach. Foehn-cold pool interactions in the Inn Valley during PIANO IOP2. Q J R Meteorol Soc. 2020 Apr;146(728):1232-1263. doi: 10.1002/qj.3735.

Adler, B., N. Kalthoff, and O. Kiseleva, 2020: Detection of structures in the horizontal wind field over complex terrain by horizontal coplanar Doppler lidar scans. Meteor. Z., doi:10.1127/metz/2020/1031.

Adler, B., A. Gohm, N. Kalthoff, N. Babic, U. Corsmeier, M. Lehner, M.W. Rotach, M. Haid, P. Markmann, E. Gast, G. Tsaknaki, G. Georgoussis, 2020: CROSSINN – a field experiment to study the three-dimensional flow structure in the Inn Valley, Austria. – Bull. Amer. Meteor. Soc. under review.
* * *

---

## Referee Comment (RC2) · Anonymous Referee #2 · 23 Sep 2020

This study presents measurements from a recent field campaign in the Alaiz mountains, northern Spain. A range of sophisticated equipment, including wind profilers, sonic anemometers and masts for measuring wind and temperature profiles, is employed to probe the flow in transects across a valley between two mountain ridges. These measurements are used to characterize two flow regimes (a hydraulic jump and a valley flow stagnation situation) for orography whose combination of horizontal scale and steepness fills a gap hitherto relatively unexplored by previous field campaigns. Consisting of an essentially observational study, containing a large amount of original results, relevant for both better understanding orographic flows and the requirements for their accurate numerical modelling, this manuscript is relevant and appropriate to Wind Energy Science. The research seems sound and the manuscript is well writ-

ten and well organized. My main objection, which is nevertheless minor and will be detailed below, concerns application of known theoretical insights, developed for idealized flows, to the measured hydraulic jump. I think that the paper should be acceptable for publication after minor revisions.

Main point

Use of the Froude number to diagnose super-critical or sub-critical flow, and therefore the occurrence of a hydraulic jump, is one of the most questionable aspects of the results presented. It is tricky to apply concepts developed for idealized cases to realistic conditions, but it must be recognized that by suitably choosing either the definition of Fr or values of the parameters included in it, Fr may vary within a wide range. D is defined as 500 m, based on the elevation difference between MP5 and M7. This seems a bit arbitrary, since D is defined by Rotunno and Lehner (2016) as the depth of a stratified layer and, for example, in Figs. 11 and 12 the atmospheric layer with stronger stratification seems much shallower. It makes more sense to equate D with the layer of high wind speed from the lidar measurements (as the authors do), but the connection with stratification still needs to be clarified. It is mentioned that the potential temperature gradient used to calculate the N included in Fr is obtained by linear interpolation between measurements at 2 and 80 m or 2 and 113 m, but this depth is much smaller than 500 m, so this value of N cannot be considered representative of the stratified layer D. I could not find any allusion to how the value of U included in Fr is estimated. All of these aspects need to be justified in a physically more convincing way, as the estimated value of Fr is very sensitive to them.

Minor points

Page 1, line 19: "several processes are included in the former but not in the latter" (when referring to meso- and micro-scale models). It would be good to be more specific here by briefly specifying what some of these processes are.

Page 3, caption of Figure 1: It should be mentioned in this caption that in the insets

the blue colour represents low elevation and yellow represents high elevation. Is the colour scale arbitrary, or does it have some quantitative meaning? It is strange that the orography of T-REX is mostly blue.

Page 4, line 3: "northwestern part of Spain". Given that these mountains are near the Pyrenees, this should be "northeastern" instead, I think.

Page 4, caption of Figure 2: "CP" is denoted by light blue, but this colour (without a nearby white colour for comparison) looks rather white instead. Consider using a different colour, or a different description (but this is just a suggestion).

Page 6, line 9: "hectometer scale". If I am not mistaken, a hectometer is 100 m. It would perhaps be easier for the reader to understand if this was phrased as "100 m" instead of "hectometer".

Page 7, line 24: "equally distributed". "equally spaced" might be a more precise description.

Page 8, line 9: "(see figure 4)". It seems to me that the technical aspects about the RHI scan that are discussed in this passage are somewhat unrelated to the aspects that are depicted in Figure 4. Consider whether it makes sense to cite that figure here.

Page 8, line 14: "staring at M7's 80 m 3D sonic storing". Although I am unfamiliar with the terminology of field work, I wonder if the word "staring" is the most technically accurate one in this context. Please check.

Page 9, Figure 5: In the description of this figure within the text, information should be included on what the angular interval is for calculating the histograms that make up these wind roses. It should also be mentioned in the caption what the percentages labelling the dotted circles mean.

Page 9, lines 12-15: "the wind rose measured at the valley floor" is described. Above, when the measurements at the mountain top are described, a turbulence intensity of 7% is mentioned (line 6). If possible, the authors should say what the turbulence

intensity in the valley floor measurements is.

Page 10, lines 4-5: "Results show that the NEWA-WRF simulations underestimate the mean wind by more than 1.5 m/s, which is indicative of unresolved speed-up effects in the meso-scale model" (a similar comment is made in page 19, lines 4-6). Do the authors believe that this underestimation is simply caused by insufficient resolution, or are there other factors at play?

Page 12, Figure 7: I am puzzled by the order in which the stability categories "nu" and "u" appear in the graphs. It would be natural to expect a sequence "u-nu-n-ns-s" in order of increasing stability, but it is "nu-u-n-ns-s" instead. Why is that? If this was a mistake, please consider correcting it (although this, of course, does not affect the correctness of the results).

Page 13, line 5: "Fr = pi U/2 N D". I am aware of the fact that the authors define the Froude number based on Rotunno and Lehner (2016), but it would be good to briefly explain the presence of the factor pi/2 in this definition.

Page 13, lines 15-16: The description of the hydraulic jump could be improved. It should be mentioned that the discontinuity that corresponds to the hydraulic jump occurs as a downstream consequence of the flow transition between subcritical and supercritical over the mountain, associated with a downslope windstorm. Perhaps a reordering of the text would do the job, since in lines 18-19 below high amplitude mountain waves and downslope flow are mentioned, and in lines 21-22, supercritical and subcritical flow regimes are mentioned.

Page 13, lines 17-18: "It differs from atmospheric gravity waves or lee waves since it involves a discontinuity and requires nonlinear dynamics to be described". Some gravity waves and lee waves also require nonlinear dynamics to be described. It is just that the nonlinearity of hydraulic jumps is more extreme.

Page 13, lines 23-24: The authors refer here to solutions that use "a Fr scaled with the

maximum wave speed given by (g D)ˆ(1/2)". Clearly this is different from the Froude number defined previously and used in the present study. How do the conditions in which this alternative Fr apply differ from those considered in the present study?

Page 13, line 30: "We defined D ∼ H ∼ 500 m in section 3.2, which agrees with previous studies". Unless the flow is the same, it is irrelevant whether this value agrees with previous studies or not, since the height of the stable layer, or whatever is used to estimate D, varies between different flows.

Page 15, lines 3-4: "Fr at MP5 decreases from 1.5 to 1.2 and the M2 measurements evidence a recirculation zone in the lee-side of the mountain". This recirculation zone is usually called a rotor, and it would be worth mentioning that term, perhaps in connection to citation of one or two studies that address that flow structure.

Page 15, line 13: "quiescent easterly winds". "Quiescent" is often reserved for situations of calm. Do you mean to say that the easterly winds are "weak", perhaps?

Page 15, lines 14-15: "The potential temperature profiles (Figure 11c) from S1, M5 and sodar RASS agree and show a stable boundary layer". What is the height of this boundary layer? What is its relation (if any) with the height of the so-called "stable layer".

Page 16, inset in Figures 11(a), and page 18, inset of Figure 13(a): These insets (containing the legends for the 2m wind speed, 2m temperature and temperature difference between 2 m and 0.36 m need to be described in the caption.

Page 16, lines 5-6: "heterogeneous land cover, which causes unequal heat fluxes". How important is this effect compared with orographic effects (causing katabatic and anabatic circulations)?

Page 17, line 8: "Both layers are stably stratified, although with different intensity". Curiously, the apparently katabatic flow shown in Figure 12 extends over a substantially higher depth than the more intensely stably stratified layer in the theta profile. Is there

an explanation for this?

Page 17, line 12: "The S7 and S9 positions show a recirculation zone". Does this correspond to pooling of the flow? If yes, it would be worth mentioning this explicitly, since cold air pools in valleys are an active area of research.

Page 18, line 2: "An elevated thermal inversion around 720 m asl". It would be good to refer back to Figure 12(b) at this point, so that readers can understand what is being discussed.

Page 18, lines 3-4: "depth of the red band within the valley observed in the RHI scan of 3:10Z". Again, it would be useful to the reader if the authors referred back to Figure 12(a), where this feature can be identified.

Page 18, lines 11-12: "ALEX17 yet poses a further challenge in numerical modeling efforts for its large domain size". I suggest replacing the word "for" in this sentence by "because of".

Page 18, line 14: "wind direction offsets up to 90°". I suggest replacing this passage by "the wind rotates by up to 90°".

Page 19, lines 1-2: I suggest that the reference at the end of the sentence spanning these two lines is moved to after "experimental report", since it seems to refer to that report.

Page 19, lines 13-14: "On the other hand, during stratified conditions with lower wind speeds, valley winds become decoupled from the mountain flow aloft due to thermal stratification". This should be connected to cold-air pooling and valley inversions, and one or two relevant references on these topics should be added.

Page 19, line 17: "in terms of its physical description". "its" should be replaced by "their", since the word it refers to, "flows", is plural.

Page 19, line 18: "i.a.". I interpret this as meaning "inter alia". Is this abbreviation

standard? If not, write it in full form instead.

Page 19: The "Author contributions" and "Competing interests" sections seem to be in the wrong place, between the Appendix title and its content. They should probably come, either before the Appendix title, or after the Appendix.

Page 21, line 22: Please check if the volume number is correct, as it appears to coincide with the year (2020).

Page 22, line 8: "pp. 1-33". I believe the "pp." is unnecessary. Consider removing it.

Page 22, line 27: The page range appears to be missing from this reference.

Page 23, lines 21-22: This reference has no volume number or page range.

Page 24, line 6: This reference has no volume number or page range.

---

## Author Comment (AC1) · 2 Nov 2020

The detailed reply to the two reviewers and one comment are found in the supplement file together with a modified version of the manuscript where the changes are highlighted.

On behalf of the authors, Pedro Santos

Please also note the supplement to this comment:
https://wes.copernicus.org/preprints/wes-2020-89/wes-2020-89-AC1-supplement.pdf

—————————————————

---

## Author Response (AR1)

**Responses to reviews of "The Alaiz Experiment: untangling multi-scale stratified flows over complex terrain" by Santos et al. wes-2020-89**

Pedro Santos

November 2, 2020

We have received three comments and reviews which are addressed in chronological order. All changes in the manuscript are clearly marked in the attached version.

**1   Review by anonymous Referee #1**

*I enjoyed reading the manuscript. It brings up very interesting flow situations from a modern measurement campaign. It is my impression that the scientists behind the study made a great effort compiling data from, and running, such an extensive campaign. In my judgement the study would be improved by clarifying the aim of the study; Is it to present the campaign? Is it to present important flow phenomena, and in that case, why are they important, and why were these two cases selected? Based on the aim of the study, the background could be improved by focusing on the previous research related to the aim. The basis of distinguishing the hydraulic jump from a lee wave could be strengthened, especially with the RASS showing semi-layered structure in the potential temperature. A linear stability analysis could potentially be useful. I think the climatological consequences of the findings could be highlighted more. The assessment of the prevalence of conditions linked to hydraulic jump was interesting, and it would be valuable with something similar also for the layered flow.*

Thanks for your comments. The objective of this paper is twofold: introduce the experiment and highlight two flow cases selected from a catalogue of flow patterns of interest, both related to wind energy and mountain meteorology. A more clear statement of these two objectives was made by rephrasing a sentence in the introduction, P. 3 L. 21–22.

The distinction between the hydraulic jump and lee-wave activity was based on the Fr number transition combined with the multi-lidar observations, from which we have identified characteristics of a hydraulic-jump-type rotor, i.a., a downslope wind separation at the valley floor and a downstream rotor extending beyond the mountaintop level. We have clarified this discussion in the paragraph stating on P.14 L.32. Yet, the layered flow case was identified mainly through the multi-lidar observations. A statistical assessment of such flow pattern could be performed applying tools such as self-organizing maps to the lidar scans, which go beyond the scope of this work.

*Abstract L. 2. Although not necessary, temporal information is also interesting.*

The temporal information is not mentioned since the acquisition frequency of the measurement equipment is many times larger than the time scale of the flow patterns. We have clarified this in the abstract.

*Abstract L.4 For which of the cases, both? That is not clear at this point.*

Yes, it is for both flow cases indeed. We have clarified the sentence.

*Abstract L.5. Here the reader does not yet know that you are referring to one of your two cases and that the reference to other cases is interesting in terms of similarity.*

We have clarified the sentence specifying that the Froude number analysis is related to the hydraulic jump case.

*Abstract. L.9 Perhaps add something on why you think it is well captured by the lidars. So far the reader has only been introduced to the lidar measurements (no other measurements), which means that the flow disconnect could potentially be a measurement problem.*

We have clarified the sentence mentioning other ground-based observations.

*Introduction L1. Does this refer to climatological or time series? For climatological, what is the basis for saying 4% is low? I understand it may be low compared to historical perspective, but in terms of economy it is perhaps not such a low uncertainty*

It refers to the climatological mean. The 4% is mentioned as low in a historical perspective. We have clarified the sentence with reference to points mentioned by the reviewer.

*P1. L.15. Additional suitable references for this claim. For example Ayotte 2008 for micro scale models, recent work on validation for the NEWA model (perhaps Drenkämper 2020). In terms of analytical work there are also investigations into this by Finnigan and Belcher (2004), and perhaps also the original theory for flow over hill by Hunt.*

We have rephrased the sentence with a more quantitative claim referencing the NEWA validation paper by Dörenkämper et al., as suggested.

*P1. L.20 As far as I remember the cited literature does not give a summary of the available validation data. I do not so much argue with the claim, but I'm not convinced the cited literature really backs up the claim. In fact, due to the NEWA project, there are quite a few data-sets being produced, so perhaps new data-sets are not so rare anymore. I do think the cited literature provides a good reference for the need of validation data.*

This paper has the goal of presenting the Alaiz experiment, which is in fact the last full-scale NEWA experiment. Therefore, it aims to fill a gap of high-quality datasets with multiple ground-based observations in complex terrain. We have replaced the word "rare" to "scarce" and added Mann et al. (2017) as a complementary reference.

*P2. L.16. Please be specific when referring to new insights.*

We have rephrased the sentence stating the objectives of the referenced studies.

*P2. L34 For sure, with such a steep a profile, a linear model is bound be somewhat inaccurate. However, as far as I remember, the performance of the non-linear models was not altogether convincing, which might also be worth mentioning.*

We agree and we have rephrased the sentence pointing to the biases of non-linear results.

*P2. L34. As far as I recall, the Bechmann et al (2011) only used neutral conditions. Are you referring to Berg et al. (2011)? In that case variation in flow pattern or something similar might be better, since I don't think there were any flow model validation in that study, and hence errors might be misleading*

Yes, this claims refers to Berg et al. (2011). We have rephrased the sentence accordingly and added the reference for clarity.

*P4. L4. Is there a reference for the airborne laser scans?*

Yes, we have added a reference from the experiment's dataset.

*P.5 L.16. I don't think only WLS70 is enough information to characterize the lidar. Consider also including the brand name.*

We agree and we have added details of the manufacturer.

*P5 L.24 I suppose the availability of the WLS70 is also interesting if it is reported for the tower. Tower width and boom lengths are good to report in my view, as they provide good indication on flow distortion magnitude. Also, data acquisition rates and averaging periods are good to report here if they are not reported further down the text.*

These information are stated in the experiment campaign & data report (Cantero et al., 2019), now referenced in the beginning of the section as suggested by the reviewer in the comment below.

*P6. L2. What type of sonic? And also same as above. Boom length and tower widths are potentially interesting.*

Brand and model of the sonic anemometers are now stated. Further information is in the experiment campaign & data report.

*P6. L.13. Reference for the RASS system is missing.*

Details of the manufacturer are now included.

*P7. L3. Consider moving this reference to the beginning of the instrumentation section.*

We have moved as suggested, since we agree that the reference to the experiment campaign & data report in the beginning of the section will be helpful for the reader that wishes to directly have a more detailed description of the instrumentation.

*P7 L10. Consider also mentioning what maximum dz difference this implies. On the other hand, I suppose also errors in the azimuth angle are interesting.*

We have complemented the sentence with the azimuth and elevation error at a reference line-of-sight distance of 4000 m. Both azimuth and elevation pointing errors are stated in the referenced table.

*P8 L10. From this I take it you first average the radial wind speeds and then compute horizontal vectors? This could be clarified in the beginning of the paragraph, as I suppose the option of calculating horizontal wind vectors and then averaging also exist. The earlier mentioning synchronization makes it a bit unclear.*

No, the RHI scans are superimposed meaning that each one depicts radial wind speeds, i.e., along the laser beam. The coplanar RHI scans give the possibility to reconstruct the wind vector projected onto the transects plane within the overlapping region of both scans, in the middle of the valley. Here, we choose only to superimpose the scans since the studied flow phenomena happens in a spatial scale that covers the entire scanning area.

*P10 L6. "Can represent the wind climate" is perhaps a bit general. I see what the authors mean, but on the other hand even 8 years leave room for some uncertainty regarding the true wind climate (in terms of for example the true long term average wind speed). In terms of containing a representative set of meteorological conditions I agree with the authors.*

We agree that the claim is overstated and we have rephrase it accordingly.

*P10 l.14 As in 20% of the slopes higher than 16.7%?*

The sentence was rephrased for clarity.

*P10, final line: Which is defined here as ...*

Ok, modifiied.

*P11 L11. The reference height of 10 m is rather low for the scales of this paper, I suppose? The scales in Berg et al (2011) were much smaller. This means the stratification at levels relevant to wind turbines will generally be much stronger, since those heights are a magnitude larger. I think it is useful also to use 10 m, but some discussion might be warranted. Did the authors try to use Ri number from the RASS to assess large scale stratification?*

We have performed a complementary analysis using a gradient Richardson number, where the potential temperature gradient has been taken from the RASS and the wind shear computed using multiple heights as in Högström (1988), but with a higher-order polynomial. This analysis gave similar results in terms of distribution of stability classes as well as mean vertical wind speed profiles. We have added a sentence stating that this analysis was performed.

*P11 L15. Are the wind speeds also at 10 m height? The same point as above could be made here as well. One could argue that for wind turbine applications the dependence of wind speed with stratification should be representative for turbine heights. Fig. 7 may give the impression that for higher wind speeds it is almost exclusively neutral conditions, while for wind turbine relevant heights the situation might be different.*

We agree that the usage of Obukhov length close to the surface is representative to a limited size of eddies in the atmosphere. However, the objective of section 3.2 was to solely present a picture of the distribution of wind conditions in terms of stratification at the valley center. For the stratified periods where the selected large-scale flow patterns are studied we have chosen to use the Froude number.

*Figure 7. Since the experiment was in Spain, it might be good to mention what time is referred to for the x-axis. Local (sun) time, local (Spain) time, or UTC? Why does NU come before U?*

The time is in UTC and this information was added in the caption. The legend with stability classes was corrected.

*Figure 8. In my opinion the figure would be improved by adding confidence bounds and possibly lines connecting the dots. It would also be valuable to know the filtering criteria for the averaging, or at least which period that is used.*

We agree and Figure 8 was redone, with shaded areas denoting associated errors of the mean. The normalized mean profiles represent the entire extensive measurement period, hence in the text we have also added the number of 30-min profiles considered for each analyzed sector.

*P.11 L26 Move this to P13 l 10, or move what is there up here (what you use as reference temperature).*

Agreed and the sentences were rearranged.

*P13. L15. Reference for this statement? I rather thought of a hydraulic jump as a continuity phenomenon (including a transition to turbulence, linked to the lower velocity), with subsequent increase of potential energy. But I'm not that familiar with theory. I guess I object to that the atmosphere "wants" to recover potential energy.*

Kaimal and Finningan (1994) state "The basic categories of supercritical flow (where the inversion rises over the hilltop) and subcritical flow (where it drops) as well as the possibility of hydraulic jumps, the abrupt, turbulent transitions from one state to another [...]". We have replaced the word "discontinuity" to "abrupt and turbulent transtion" for clarity and added this reference.

*Figure 9, and accompanying discussion. Do you have any criteria for distinguishing a hydraulic jump from a regular lee wave? The finite length of the low velocity region behind the jump in Fig. 9 (particularly in panel 2) makes it look to me also a bit like a wave (also the shape of the jet in panel 1).*

The characterization of the hydraulic jump in this work is based on: (i) the observed supercritical to subcritical transition, (ii) lidar measurements showing that the downslope flow separation occurs at the valley floor and (iii) the lidar-observed rotor downstream extends beyond the mountaintop level. However, we acknowledge that towards the end of the period these flow features are no longer observed and the observations tend to evidence a lee-wave-type rotor. We have made this clear in the text.

*Fig. 10. Confidence bounds (even if estimated) would help the argument made in the text (that the Froude number really decreases at MP5).*

We do not see how to estimate such confidence bounds. Instead, we have added a sentence discussing the intrinsic variability of the Froude computation in this study (P. 13 L. 9–11), which do not necessarily represent the entire inversion depth. Furthermore, the decrease in Fr number is connected to the change in flow behavior to a lee-wave-type rotor, as shown by the multi-lidar measurements.

*P15. L18. Horizontal scale?*

Agreed and rephrased.

*Fig. 12. Did you identify the cause of the layers with reverse (negative) wind speed? It is interesting to me that they are on the downstream mountain side. Is it drainage currents from the mountain side?*

During this case, the northern general wind is channelized along the Elorz valley generating a westerly flow. As the evening advances and the surface thermal inversion grows, this western flow takes either a more northern (blue color) or southern (red) component depending on the location within the valley atmosphere. The red areas in Figure 12 indicate those locations of the valley were the flow has a southwesterly direction, mainly located at the valley center and over the southern slope. The cause for this wind direction could not be fully identified. It could either have a dynamic origin (i.e. the stably stratified flow enters into the valley with a northwesterly direction and turns into a southwesterly direction after impacting over the southern slopes of the valley) or be the result of a westerly wind merged with a southern drainage flow generated over the mountain southern slopes.

*P18 L5. I think in order to say that this study presents the Alaiz experiment, there should be more substance regarding the experimental details. Perhaps introduces is a better word. Also, I think focusing on the flow patterns is interesting enough.*

We agree and the first sentence of the conclusions section was rephrased.

*Technical comments The tempus change throughout the text.*

We believe this is good practice in scientific writing. Events that occurred in the past should be described in the past tense. See `https://www.nature.com/scitable/topicpage/effective-writing-13815989/` for further explanation. For example, Section 2 contains a description of the experiment execution (past tense) whereas sections 3 and 4 show results of this study (present tense).

**2   Comment by B. Adler**

*In their introduction, the authors provide a nice overview of recent campaigns focusing on flow features in complex terrain categorized after domain size and slope gradient. However, we think that they missed several recent relevant measurement campaigns siting in the center part of their diagram in Fig. 1. We encourage the authors to consider including these campaigns to complement their overview. One is the PIANO campaign which took place in the Inn Valley over Innsbruck in Austria in 2017 focusing on cold pool and foehn interactions (Haid et al., 2020). Several scanning Doppler lidars were deployed in a complex scanning configuration consisting of coplanar RHI and PPI scans as well as vertical stare mode to get insight into the mean and turbulent structure of the three-dimensional flow. Recent coplanar Doppler lidar measurements were conducted in the complex topography around the city of Stuttgart in south-western Germany focusing on mesoscale flow structures and convective cells (Adler et al. 2020). In 2019, a set of six Doppler lidars was deployed about 20 km east of Innsbruck in the Inn Valley to study the three-dimensional flow structure in the valley under different synoptic conditions. Three of the lidars performed synchronized coplanar RHI scans to capture the cross-valley kinematic flow structure across the whole valley. A BAMS paper about the campaign and measurements is currently under review (Adler et al. 2020). Some information on the campaign were presented during the 19th Conference on Mountain Meteorology and on the KIT website.*

We thank you for the comment. We were not aware of these mentioned campaigns in the Inn Valley involving multi-lidar measurements. Therefore, we have decided to modify Figure 1 and include an insert with the Inn Valley area that covers both the PIANO and CROSSINN experiments. We have also added a reference of each experiment in the Introduction.

**3   Review by anonymous Referee #2**

*This study presents measurements from a recent field campaign in the Alaiz mountains, northern Spain. A range of sophisticated equipment, including wind profilers, sonic anemometers and masts for measuring wind and temperature profiles, is employed to probe the flow in transects across a valley between two mountain ridges. These measurements are used to characterize two flow regimes (a hydraulic jump and a valley flow stagnation situation) for orography whose combination of horizontal scale and steepness fills a gap hitherto relatively unexplored by previous field campaigns. Consisting of an essentially observational*

*study, containing a large amount of original results, relevant for both better understanding orographic flows and the requirements for their accurate numerical modelling, this manuscript is relevant and appropriate to Wind Energy Science. The research seems sound and the manuscript is well written and well organized. My main objection, which is nevertheless minor and will be detailed below, concerns application of known theoretical insights, developed for idealized flows, to the measured hydraulic jump. I think that the paper should be acceptable for publication after minor revisions.*

*Main point: Use of the Froude number to diagnose super-critical or sub-critical flow, and therefore the occurrence of a hydraulic jump, is one of the most questionable aspects of the results presented. It is tricky to apply concepts developed for idealized cases to realistic conditions, but it must be recognized that by suitably choosing either the definition of Fr or values of the parameters included in it, Fr may vary within a wide range. D is defined as 500 m, based on the elevation difference between MP5 and M7. This seems a bit arbitrary, since D is defined by Rotunno and Lehner (2016) as the depth of a stratified layer and, for example, in Figs. 11 and 12 the atmospheric layer with stronger stratification seems much shallower. It makes more sense to equate D with the layer of high wind speed from the lidar measurements (as the authors do), but the connection with stratification still needs to be clarified. It is mentioned that the potential temperature gradient used to calculate the N included in Fr is obtained by linear interpolation between measurements at 2 and 80 m or 2 and 113 m, but this depth is much smaller than 500 m, so this value of N cannot be considered representative of the stratified layer D. I could not find any allusion to how the value of U included in Fr is estimated. All of these aspects need to be justified in a physically more convincing way, as the estimated value of Fr is very sensitive to them.*

Firstly, we thank the reviewer for the comments. We agree that using a two-layer flow theory (Long, 1954) to describe a full-scale three dimensional flow has several caveats, as detailed by Strauss et al., 2016. Since the Froude number is an important parameter for the analysis, we have extended the explanation of how it is computed (including the choice for $U$) and added a discussion of the caveats that emerge as a result of our choices, including the fact that neither $U$ nor $N$ represent the entire inversion depth.

Furthermore, we would like to clarify to the reviewer that the classification of this flow episode relied mostly on the multi-lidar observations. From them, we have identified characteristics of a hydraulic-jump-type rotor, i.a., a downslope wind separation at the valley floor and a downstream rotor extending beyond the mountaintop level. We acknowledge the influence of the Tajonar ridge downstream the mountain and the presence of valley flow with channeling effects dissociate this episode from an idealized rotor assumed by the two-layer model. However, we believe this analysis adds value to the characterization of lee-side atmospheric hydraulic jumps and that the observations bear sufficient similarity to deserve this classification.

*Page 1, line 19: "several processes are included in the former but not in the latter" (when referring to meso- and micro-scale models). It would be good to be more specific here by briefly specifying what some of these processes are.*

We have rephrased and added specific examples.

*Page 3, caption of Figure 1: It should be mentioned in this caption that in the insets the blue colour represents low elevation and yellow represents high ele-*

*vation. Is the colour scale arbitrary, or does it have some quantitative meaning? It is strange that the orography of T-REX is mostly blue.*

The explanation of the color bar of the inserts is now explained in the caption and in the text. The color scale aims to represents the largest elevation changes of each site. Both T-REX and WFIP2 sites were adjusted and a new site (Inn Valley) was added, as suggested above by B. Adler.

*Page 4, line 3: "northwestern part of Spain". Given that these mountains are near the Pyrenees, this should be "northeastern" instead, I think.*

We have rephrased to "northern part of Spain" instead.

*Page 4, caption of Figure 2: "CP" is denoted by light blue, but this colour (without a nearby white colour for comparison) looks rather white instead. Consider using a different colour, or a different description (but this is just a suggestion).*

The image is in high quality, so one can zoom in and distinguish the colors.

*Page 6, line 9: "hectometer scale". If I am not mistaken, a hectometer is 100 m. It would perhaps be easier for the reader to understand if this was phrased as "100 m" instead of "hectometer".*

The expression of 'hectometer scale' refers to 'the scale of 100 m', as the reviewer points out. Thus, we do not see the need of changing it.

*Page 7, line 24: "equally distributed". "equally spaced" might be a more precise description.*

Agreed and modified.

*Page 8, line 9: "(see figure 4)". It seems to me that the technical aspects about the RHI scan that are discussed in this passage are somewhat unrelated to the aspects that are depicted in Figure 4. Consider whether it makes sense to cite that figure here*

The objective of this reference is to direct the reader to the essential part of the experimental setup which is actually used and analyzed in this study.

*Page 8, line 14: "staring at M7's 80 m 3D sonic storing". Although I am unfamiliar with the terminology of field work, I wonder if the word "staring" is the most technically accurate one in this context. Please check*

We have replaced "staring" with "aiming".

*Page 9, Figure 5: In the description of this figure within the text, information should be included on what the angular interval is for calculating the histograms that make up these wind roses. It should also be mentioned in the caption what the percentages labelling the dotted circles mean.*

The caption of Figure 5 was complemented with the suggested descriptive details.

*Page 9, lines 12-15: "the wind rose measured at the valley floor" is described. Above, when the measurements at the mountain top are described, a turbulence intensity of 7% is mentioned (line 6). If possible, the authors should say what the turbulence intensity in the valley floor measurements is.*

We added a sentence stating the average turbulence intensity at 15 m s$^{-1}$ observed from M5 at the valley floor.

*Page 10, lines 4-5: "Results show that the NEWA-WRF simulations underestimate the mean wind by more than 1.5 m/s, which is indicative of unresolved speed-up effects in the meso-scale model" (a similar comment is made in page 19, lines 4-6). Do the authors believe that this underestimation is simply caused by insufficient resolution, or are there other factors at play?*

The manuscript does not aim to identify the origins of wind speed biases between NEWA-WRF and observations at the mountain top. We are aware of a further analysis at Alaiz using WRF with a horizontal resolution of 1 km in which the mean wind speed bias at the mountain top is improved, but other factors, e.g., synoptic effects from ERA5 not corrected by WRF, cannot be ruled out. Furthermore, Dörenkämper et al. (in press) argue that NEWA-WRF's "mean wind speed biases are not systematically associated with mischaracterisation of the effective surface roughness or of internal boundary layer effects".

*Page 12, Figure 7: I am puzzled by the order in which the stability categories "nu" and "u" appear in the graphs. It would be natural to expect a sequence "u-nu-n-ns-s" in order of increasing stability, but it is "nu-u-n-ns-s" instead. Why is that? If this was a mistake, please consider correcting it (although this, of course, does not affect the correctness of the results).*

The legend was corrected.

*Page 13, line 5: "Fr = pi U/2 N D". I am aware of the fact that the authors define the Froude number based on Rotunno and Lehner (2016), but it would be good to briefly explain the presence of the factor pi/2 in this definition*

Done.

*Page 13, lines 15-16: The description of the hydraulic jump could be improved. It should be mentioned that the discontinuity that corresponds to the hydraulic jump occurs as a downstream consequence of the flow transition between subcritical and supercritical over the mountain, associated with a downslope windstorm. Perhaps a reordering of the text would do the job, since in lines 18-19 below high amplitude mountain waves and downslope flow are mentioned, and in lines 21-22, supercritical and subcritical flow regimes are mentioned*

We have reordered one sentence in the paragraph to highlight the hydraulic jump connection with a downslope flow in the lee side of the mountain. The further explanation of supercritical to subcritical flow transition remained in the second paragraph since we consider it needs a proper reference and explanation, in connection with the Froude number, for the wind energy community readers that might not be familiar with such concepts.

*Page 13, lines 17-18: "It differs from atmospheric gravity waves or lee waves since it involves a discontinuity and requires nonlinear dynamics to be described". Some gravity waves and lee waves also require nonlinear dynamics to be described. It is just that the nonlinearity of hydraulic jumps is more extreme*

Agreed. We have rephrased the sentence to clarify that the hydraulic jump discontinuity is stronger.

*Page 13, lines 23-24: The authors refer here to solutions that use "a Fr scaled with the maximum wave speed given by (g D)^(1/2)". Clearly this is different from the Froude number defined previously and used in the present study. How do the conditions in which this alternative Fr apply differ from those considered in the present study?*

The Fr number computation of this study is based on observations up to 80 m above ground from meteorological masts, therefore they are local and do not necessarily represent the entire stratified layer. We have clarified such caveats in the end of section 3. The scaling using $\sqrt{g'D}$, where $g'$ is a reduced gravitational acceleration, could also be applied here if we were able to compute atmospheric variables throughout the boundary layer, e.g., using outputs from numerical simulations.

*Page 13, line 30: "We defined D ≈ H ≈ 500 m in section 3.2, which agrees with previous studies". Unless the flow is the same, it is irrelevant whether this value agrees with previous studies or not, since the height of the stable layer, or whatever is used to estimate D, varies between different flows*

The mentioned study also assessed the Froude number at the MP5 location, however we agree with the reviewer in the sense that the selection of $D$ is arbitrary. Therefore, we have removed the reference and rephrased the sentence accordingly.

*Page 15, lines 3-4: "Fr at MP5 decreases from 1.5 to 1.2 and the M2 measurements evidence a recirculation zone in the lee-side of the mountain". This recirculation zone is usually called a rotor, and it would be worth mentioning that term, perhaps in connection to citation of one or two studies that address that flow structure.*

We have rephrased the sentence mentioning that, towards the end of the episode, M2 is located at the bottom of the hydraulic jump rotor, where reverse surface winds are observed. The observations made by Strauss et al. (2016) during the T-REX experiment are referenced here.

*Page 15, line 13: "quiescent easterly winds". "Quiescent" is often reserved for situations of calm. Do you mean to say that the easterly winds are "weak", perhaps?*

Agreed. 'Quiescent' has been changed into 'weak'.

*Page 15, lines 14-15: "The potential temperature profiles (Figure 11c) from S1, M5 and sodar RASS agree and show a stable boundary layer". What is the height of this boundary layer? What is its relation (if any) with the height of the so-called "stable layer".*

In this section we were not intended to analyse the boundary layer height and its relation with the temperature profile within the valley. The expression of a "stable boundary layer" was used here to emphasize that the thermal inversion depicted in figure 11c indicates the presence of a stably stratified layer over the valley floor. Since this emphasis may confuse some readers, we have modified it with a more precise description: "The potential temperature profiles (Figure 11c) from S1, M5 and sodar RASS agree and show a thermal inversion".

*Page 16, inset in Figures 11(a), and page 18, inset of Figure 13(a): These insets (containing the legends for the 2m wind speed, 2m temperature and temperature difference between 2 m and 0.36 m need to be described in the caption*

Done.

*Page 16, lines 5-6: "heterogeneous land cover, which causes unequal heat fluxes". How important is this effect compared with orographic effects (causing katabatic and anabatic circulations)?*

Both 'mountainous terrain' and 'heterogeneous land cover' are equally important. We have slightly modified the sentence to explicitly mention both effects.

*Page 17, line 8: "Both layers are stably stratified, although with different intensity". Curiously, the apparently katabatic flow shown in Figure 12 extends over a substantially higher depth than the more intensely stably stratified layer in the theta profile. Is there an explanation for this?*

During this study case, the northern general wind is channelized along the Elorz valley generating a westerly flow. Two clear stages characterise this night: 1) a moderate general wind that interacts with the valley atmosphere until

midnight and 2) the decoupling between the valley atmosphere and the overlying flow as it diminishes intensity after midnight.

These two distinct stages are described in paragraph 3-4 and 5, respectively. Thus, the different stable layers mentioned in paragraph 3 correspond approximately to the red and blue bands that appear in the cross sections of 22:10Z and 00:10Z in figure 12a. We agree that the red band grows vertically covering a great part of the less stable layer at 03:10Z. We believe that this is due to the decoupling between the valley atmosphere and the general wind, as explicitly mentioned in paragraph 5 of this section.

Therefore, for clarification, we have included the times of the RHI scans that we refer to in paragraph 3.

Finally, we'd like to clarify to the reviewer that the red areas in Figure 12 indicate those locations of the valley were the flow has a southwesterly direction, mainly located at the valley center and over the southern slope. The cause for this wind direction could not be fully identified. It could either have a dynamic origin (i.e. the stably stratified flow enters into the valley with a northwesterly direction and turns into a southwesterly direction after impacting over the southern slopes of the valley) or be the result of a westerly wind merged with a southern drainage flow generated over the mountain southern slopes. In any case, this wind dynamics occurs above the stronger thermal inversion developed over the valley floor after midnight. Other works have reported the existence of drainage flows generated over sloping surfaces that flow over the coldest air accumulated at the bottom of the valley (Cuxart et al. 2007, Martínez and Cuxart, 2009).

*Page 17, line 12: "The S7 and S9 positions show a recirculation zone". Does this correspond to pooling of the flow? If yes, it would be worth mentioning this explicitly, since cold air pools in valleys are an active area of research.*

We agree that cold air pooling in valleys is a very interesting and active area of research. The weak wind speeds observed in S7 and S9 can be explained due to the sheltering effect of the northern mountain range, favouring a strong thermal inversion at the skin level. Thus, 'recirculation zone' has been replaced by 'sheltered zone'. However, we do not feel appropriate to include the 'cold air pool' concept here because these stations are located over the mountain slope (S9) and at the foothills (S7). Nevertheless, the formation of a cold air pool over the valley floor after midnight is explicitly mentioned in the last paragraph of this section.

*Page 18, line 2: "An elevated thermal inversion around 720 m asl". It would be good to refer back to Figure 12(b) at this point, so that readers can understand what is being discussed*

Done.

*Page 18, lines 3-4: "depth of the red band within the valley observed in the RHI scan of 3:10Z". Again, it would be useful to the reader if the authors referred back to Figure 12(a), where this feature can be identified*

Done.

*Page 18, lines 11-12: "ALEX17 yet poses a further challenge in numerical modeling efforts for its large domain size". I suggest replacing the word "for" in this sentence by "because of"*

Agreed and sentence was rephrased.

*Page 18, line 14: "wind direction offsets up to $90°$". I suggest replacing this passage by "the wind rotates by up to $90°$"*

Agreed and sentence was rephrased.

*Page 19, lines 1-2: I suggest that the reference at the end of the sentence spanning these two lines is moved to after "experimental report", since it seems to refer to that report.*

Agreed and reference was moved.

*Page 19, lines 13-14: "On the other hand, during stratified conditions with lower wind speeds, valley winds become decoupled from the mountain flow aloft due to thermal stratification". This should be connected to cold-air pooling and valley inversions, and one or two relevant references on these topics should be added*

An explicit mention to the formation of a cold air pool, together with a couple of references, has been added into the sentence.

*Page 19, line 17: "in terms of its physical description". "its" should be replaced by "their", since the word it refers to, "flows", is plural*

Agreed and sentence was rephrased.

*Page 19, line 18: "i.a.". I interpret this as meaning "inter alia". Is this abbreviation standard? If not, write it in full form instead.*

Yes, it stands for "inter alia" and we consider understandable for the reader without the full form.

*Page 19: The "Author contributions" and "Competing interests" sections seem to be in the wrong place, between the Appendix title and its content. They should probably come, either before the Appendix title, or after the Appendix*

The sections were rearranged.

*Page 21, line 22: Please check if the volume number is correct, as it appears to coincide with the year (2020).*

The reference was updated.

*Page 22, line 8: "pp. 1-33". I believe the "pp." is unnecessary. Consider removing it.*

The reference was updated.

*Page 22, line 27: The page range appears to be missing from this reference.*
The reference was corrected.

*Page 23, lines 21-22: This reference has no volume number or page range.*
This reference represents a dataset.

*Page 24, line 6: This reference has no volume number or page range.*
The reference was corrected.

**The Alaiz Experiment: untangling multi-scale stratified flows over complex terrain**

Pedro Santos[1], Jakob Mann[1], Nikola Vasiljević[1], Elena Cantero[2], Javier Sanz Rodrigo[2], Fernando Borbón[2], Daniel Martínez-Villagrasa[3], Belén Martí[3], and Joan Cuxart[3]

[1]Technical University of Denmark, DTU Wind Energy, Risø Campus, Roskilde, Denmark
[2]National Renewable Energy Centre (CENER), Sarriguren, Spain
[3]Physics Department, University of the Balearic Islands (UIB), Mallorca, Spain

**Correspondence:** Pedro Santos (paas@dtu.dk)

**Abstract.** We present novel measurements from a field campaign that aims to characterize multi-scale flow patterns, ranging from 0.1 to 10 km in a time-resolved manner, in a mountainous region in Northwestern Spain with a mountain-valley-ridge configuration. We select two flow cases where topographic-flow interactions were measured by five synchronized scanning Doppler wind lidars along a 10-km transect line  that includes a cross-section of the valley. We observed a hydraulic jump in the lee-side of the mountain.  For this case, the Froude number transition from supercritical ($> 1$) at the mountain to subcritical ($< 1$) at the valley is in agreement with previous experiments at a smaller scale. For a one-year period, the measurements show such a transition about 10 % of the time, indicating a possible high occurrence of hydraulic jumps. The second flow case presents valley winds that are decoupled from the northerly flow aloft and show a stratified layered pattern, which is well captured by the lidar scans and complementary ground-based observations. These measurements can aid the evaluation of multi-scale numerical models as well as improving our knowledge with regards to mountain meteorology.

**1   Introduction**

Over flat and homogeneous terrain, such as areas far offshore, the difference between measured and simulated climatological mean wind speeds  at wind energy relevant heights  is in some cases less than 4% (Olsen et al., 2017). This is historically low although there is still economic value in reducing it even further. However, over complex terrain, with steep slopes and varying land cover, such differences can be closer to 10% (Dörenkämper et al., 2020) depending on terrain complexity, implying large uncertainties on the estimated annual energy production of wind farms. Even small deviations in the terrain description over a given area may result in substantial differences in the simulated flow (Lange et al., 2017).

For the prediction of winds in complex terrain, meso-scale models, typically covering scales down to a kilometer or so, have to be coupled with micro-scale models that cover smaller scales down to meters. Meso- and micro-scale models are fundamentally different in the sense that flow processes that are parameterized in the former are resolved in the latter, while several physical processes, e.g., cumulus clouds and convective systems, are included in the former but not in the latter. The scales that are at the interface of the two models have been dubbed *terra incognita* by Wyngaard (2004) and this experimental

investigation aims to explore some sub-meso-scale physical processes. New datasets from complex terrain experiments with details on flow patterns covering these scales are  scarce and needed to evaluate and quantify the uncertainty of numerical models (Mann et al., 2017; Sanz Rodrigo et al., 2017). Apart from wind energy, untangling flow over complex terrain is of general interest for the mountain meteorology community (Serafin et al., 2018).

Over the last decades, experimental efforts have been conducted with increasing density of instruments and types of measurement aiming to better understand flow conditions in hilly and mountainous terrain. A well-known experiment was performed at the Askervein Hill, which became the main reference in the development and validation of pioneering analytical and linearized flow models dealing with gently sloping terrain (Salmon et al., 1988; Walmsley and Taylor, 1996). Furthermore, the Cooper's Hill experiment (Coppin et al., 1994) used meteorological masts and sonic anemometers to study the flow over a ridge as a function of atmospheric stability.

In more recent endeavors, Doppler wind lidars and airborne instrumentation have been used to characterize large-scale phenomena over steep hills and mountain ranges. Two examples of such are the terrain-induced rotor experiment (T-REX, Grubišić et al., 2008) and the mountain terrain atmospheric modeling and observations program (MATERHORN, Fernando et al., 2015). T-REX focused on low-level vortices formed downstream of a mountain ridge and MATERHORN was a multidisciplinary initiative to approach large-scale atmospheric phenomena in complex terrain, where two major experimental campaigns studied thermally driven winds with strong synoptic forcing. Back to smaller scales, detailed scanning lidar and turbulence measurements were performed at the escarpment of Bolund (Lange et al., 2016; Berg et al., 2011), which  detailed turbulence characteristics under flow-terrain interaction. A blind test followed to compare a wide variety of flow models (Bechmann et al., 2011) and wind tunnel prototypes (Kilpatrick et al., 2016; Conan et al., 2016; Lange et al., 2017).

With an extensive collaboration effort in the pursuit of new insights on wind resource characterization, a range of experiments, both onshore and offshore, were performed within the New European Wind Atlas (NEWA) project to evaluate meso- and micro-scale models (Mann et al., 2017). The experiments made extensive use of a recently developed infrastructure that uses synchronized measurements from multiple lidars, the so-called long-range WindScanner system (Vasiljević et al., 2016). The Kassel experiment, performed at the forested hill Rödeser Berg in Germany, was used to quantify the accuracy in the reconstruction of the wind vector with distinct multi-lidar combinations and the lidar's spatial averaging effect on the turbulence spectra (Pauscher et al., 2016). A methodology for the execution of experiments involving multi-lidars was developed during the double-ridge Perdigão experiment in Portugal (Vasiljević et al., 2017), which is the largest experimental venture in complex terrain to date in terms of density of measurement equipment (Fernando et al., 2019). In parallel to the NEWA project, the second Wind Forecast Improvement Project (WFIP2), also deployed a large array of instruments to cover the area around the Columbia Gorge in the United States (Wilczak et al., 2019). This experiment was also focused on the improvement of meso- and micro-scale coupling methods (Haupt et al., 2019).

The Inn Valley, located close to Innsbruck, Austria, is a site where extensive field campaigns took place to characterize atmospheric processes with regards to mountain meteorology. Recent experiments used multiple wind lidars to obtain flow

patterns to characterize cold-air pool erosion by downslope mountain winds (Haid et al., 2020) as well as cross-valley circulation cells using coplanar multi-lidar observations (Adler et al., 2020).

Figure 1 puts the Alaiz experiment in perspective of these complex terrain experiments by comparing the area covered and steepness of the terrain quantified by the upper 10th percentile of the slopes. The color in the inserts covers the elevation range of each site. In this context, Askervein can be seen as a departure point that gave rise to experiments in larger areas and steeper slopes.

With a small domain but a steep escarpment, Bolund left the realm of gentle slopes, and hence emphasized the limitations of linearized flow models and eventual biases of non-linear simulations. Due to the very small scales, neglecting the effects of atmospheric stability did not lead to major  variations in the flow pattern over Bolund (Berg et al., 2011). On the other hand, in METCRAX II, scanning lidars captured atmospheric hydraulic jumps and cool pool events inside a meteorite crater in Arizona (Lehner et al., 2016). In Kassel and Perdigão, larger areas were investigated that required the use of long-range WindScanners. Perdigão presents a double-hill configuration, 1.5 km apart, which is dominated by micro-scale effects, such as valley winds and recirculation zones, but is also affected by thermal stratification effects that can lead to internal atmospheric gravity waves under stable conditions (Menke et al., 2019; Palma et al., 2019). T-REX and WFIP2 are mountain range studies, too large to be fully covered by a single set of instruments, still with similar thermally stratified flows presented in this study. In the extreme of terrain complexity, the Inn Valley area hosted multiple experimental campaigns to cover such an alpine region.

[revised manuscript text omitted]

of dimensionless stability $10/L$ at M7. As expected, stably stratified conditions prevail at night, whereas unstable conditions dominate during daytime and peak around midday. Figure 7b shows the behavior of stability with wind speed, where a prevalence of neutral conditions with increasing wind speeds is found. We have also performed a similar analysis using the gradient Richardson number with concurrent wind speed observations from M7 and potential temperature profiles from the WindRASS. This parameter showed a similar distribution of stability classes in terms of diurnal cycle and wind speed bins (not shown).

[Figure]

**Figure 7.** Frequency of $z/L$ at 10 m agl divided in stability classes per hour **(a)** and per wind speed **(b)** from measurements at M7. The stability classes are found in table 2. The time is in UTC.

Figure 8 presents the vertical profiles of the normalized mean wind speed for two prevalent wind sectors at the valley floor, namely a NW sector (300°±30°) with 3193 30-min profiles in total and a SE sector (120°±30°) with 3120 30-min profiles, both divided by stability classes. For the each vertical level, the shaded area represents the standard error of the mean given by $\pm\sigma/\sqrt{N}$, where $\sigma$ is the standard deviation and $N$ the number of observations. For the SE sector (Fig. 8a) large differences can be observed close to the surface, showing that the surface roughness' fetch can be quite inhomogeneous for this sector but also likely influenced by land-cover seasonal effects. Also, negative wind shears characterize the stable class, which are potentially caused by valley drainage flow (Serafin et al., 2018). Profiles from the NW wind sector (Fig. 8b) resemble more

flat and homogeneous conditions, with increasing wind shear with stability and a similar normalized wind close to the surface except for stable conditions.

[Figure]

**Figure 8.** Mean vertical wind profiles normalized with the friction velocity computed by the 10-min sonic anemometer ($u_{*10}$). Panel (a) shows profiles within the $120°\pm30°$ sector and panel (b) shows profiles within the $300°\pm30°$ sector based on wind directions at 10m. Shaded areas denote associated standard errors of the mean.

Furthermore, when dealing with mountain flows, the topography occupies a large portion of the ABL and hence plays a major role in the flow stratification. Disturbances in the stable atmosphere caused by the topography may generate three-dimensional flow phenomena i.a. atmospheric lee waves, rotors and hydraulic jumps (Kaimal and Finnigan, 1994). The natural frequency of these vertical oscillations is characterized by the Brunt–Väisälä frequency $N = [(g/\overline{\theta})(\partial\overline{\theta}/\partial z)]^{1/2}$ where $\partial\overline{\theta}/\partial z$ is the potential temperature gradient. The potential temperature is computed as $\theta = T + 0.0098 z_{asl}$, where $0.0098$ Km$^{-1}$ is the dry adiabatic temperature gradient. The potential temperature gradient in $N$ is calculated by linear interpolation between the measurements at 2 and 80 m for M5/M3 and between 2 and 113 m for MP5.

The Froude number (Fr) is the dimensionless parameter given by the ratio of flow inertia based on a reference upwind wind speed $U$ to the gravity forces acting on the flow (Kaimal and Finnigan, 1994). Following the linearized solution presented by

Rotunno and Lehner (2016) Fr $= \pi U/2ND$, where D is a height for the stably stratified flow layer upstream of the mountain, also called inversion depth. The $\pi/2$ multiplier comes from a derivation considering critical flow, i.e., Fr = 1. The scaling can also be performed with a characteristic mountain height H.

We  choose the inversion depth D  to be proportional to the high wind speed layer height from the lidar measurements (c.f. figure 9)  and equal to 500 m, i.e., it is an empirical selection based on this case study. The characteristic height H is considered as the elevation difference between MP5 and M7, which is ≈ 500 m, hence $H \sim D \approx$ 500 m. Fr is evaluated at the Alaiz mountain top (MP5) and foothills (M2), at the valley floor (M5) and at the north ridge slope (M3) using the same $D$.  Furthermore, the characteristic wind speed $U$ is taken at 80 m  at all positions.

It is worth noticing that the computation of Fr in this study has some caveats, as neither $U$ nor $N$ represent the entire inversion depth and D is a characteristic length with a somewhat arbitrary value. Here, our aim is to investigate the relative changes in Fr, where we compare Fr estimations at the mountain top  with the ones along the valley.

**4 Selected flow cases**

**4.1 The Lee-Side Hydraulic Jump**

[revised manuscript text omitted]

Figures 13b and c represent the wind speed and wind direction measured at MP5 118 m agl and at 80 m agl by distinct masts across the valley. There is a persistent offset of 90° in the wind direction between the mountain (MP5) and valley (M2,M3,M5), while the general wind decreases throughout the evening, as indicated by MP5. In consequence, the wind speed also diminishes within the valley, intensifying the surface cooling and generating a stronger thermal inversion at the valley floor after midnight (figure 12b). This situation favours the development of a cold air pool (Serafin et al., 2018) which decouples the surface layer, with winds responding to a local regime, from the southwesterly flow within the valley (not shown). An elevated thermal

inversion around 720 m asl  (see profile at 03:10Z in figure 12b) decouples the valley atmosphere from the northerly wind aloft, increasing the depth of the red band within the valley observed in the RHI scan at 03:10Z (figure 12a).

**5   Summary and Conclusions**

[revised manuscript text omitted]

*Competing interests.* The authors declare no conflict of interest.

10 *Acknowledgements.* We would like to thank Michael Courtney and all staff from DTU Wind Energy (Denmark), CENER and UIB (Spain) involved on the commissioning and ongoing monitoring of the experimental infrastructure. We credit Alfredo Peña for reviewing the manuscript and for the analysis that resulted in Figure 1. We would also like to thank Ebba Dellwik for her guidance during the manuscript writing, Bianca Adler for her useful comment during the open discussion and two anonymous referees for a thorough review. The European Commission (EC) partly funded NEWA project (NEWA- New European Wind Atlas, FP7-ENERGY.2013.10.1.2, European Commission's grant agreement number 618122). National agencies that also provided funding are the *Danish Energy Agency* and *Ministerio de Economia y Competitividad* (Spain). B.M. was supported by the grant FPI-CAIB (FPI/2165/2018) of the *Vicepresidència i Conselleria d'Innovació, Recerca i Turisme del Govern de les Illes Balears* and the *Fons Social Europeu*. The WindRASS and SEB station were rented from the Meteorological Service of Catalonia with funds of the research project PCIN-2014-016-C07-01.

**Appendix A: Lidar configuration for RHI transect scan**

20 ~~We would like to thank Michael Courtney and all staff from DTU Wind Energy (Denmark), CENER and UIB (Spain) involved on the commissioning and ongoing monitoring of the experimental infrastructure. We credit Alfredo Peña for the analysis that resulted in Figure 1. The European Commission (EC) partly funded NEWA project (NEWA- New European Wind Atlas, FP7-ENERGY.2013.10.1.2, European Commission's grant agreement number 618122). National agencies that also provided funding are the *Danish Energy Agency* and *Ministerio de Economia y Competitividad* (Spain). B.M. was supported by the grant~~
25

| Parameters | WS3 | WS5 |
|---|---|---|
| (Easting, Northing) [m] | (617846.80,4732496.43) m | (617305.59,4729848.75) m |
| Elevation [m asl] | 487.21 m | 545.92 m |
| Azimuth [$^o$] | 191.55° | 11.55° |
| Elevation (min,max)$^o$ | (3°,18°) | (1.58°,21.56°) |
| Scan speed [$^o s^{-1}$] | 0.5°s$^{-1}$ | 0.666°s$^{-1}$ |
| Range gates [min:$\Delta$:max] m | [100:50:5000] m | [100:20:5000] m |
| Accumulation time [ms] | 1000 ms | |
| Pulse length [ns] | 400 ns | |
| Points per range gate [-] | 128 | |
| Signal spectral width [ms$^{-1}$] | 3 m s$^{-1}$ | |
| Specified physical resolution [m] | 75 m | |

**Table A1.** Windscanner's position and configuration for the Transect RHI scans.